# Systems glycomics of adult zebrafish identifies organ-specific sialylation and glycosylation patterns

Nao Yamakawa[1,2], Jorick Vanbeselaere[1], Lan-Yi Chang[1,3], Shin-Yi Yu[1], Lucie Ducrocq[1], Anne Harduin-Lepers [1], Junichi Kurata[4], Kiyoko F. Aoki-Kinoshita[4], Chihiro Sato[2], Kay-Hooi Khoo[3], Ken Kitajima[2] & Yann Guerardel [1]

The emergence of zebrafish *Danio rerio* as a versatile model organism provides the unique opportunity to monitor the functions of glycosylation throughout vertebrate embryogenesis, providing insights into human diseases caused by glycosylation defects. Using a combination of chemical modifications, enzymatic digestion and mass spectrometry analyses, we establish here the precise glycomic profiles of eight individual zebrafish organs and demonstrate that the protein glycosylation and glycosphingolipid expression patterns exhibits exquisite specificity. Concomitant expression screening of a wide array of enzymes involved in the synthesis and transfer of sialic acids shows that the presence of organ-specific sialylation motifs correlates with the localized activity of the corresponding glycan biosynthesis pathways. These findings provide a basis for the rational design of zebrafish lines expressing desired glycosylation profiles.

[1] Université de Lille, CNRS, UMR 8576 – UGSF—Unité de Glycobiologie Structurale et Fonctionnelle, F- 59000 Lille, France. [2] Bioscience and Biotechnology Center, Nagoya University, Nagoya 464-8601, Japan. [3] Institute of Biological Chemistry, Academia Sinica, Taipei 11529, Taiwan. [4] Faculty of Science and Engineering, Soka University, Hachioji, Tokyo 192-8577, Japan. These authors contributed equally: Nao Yamakawa, Jorick Vanbeselaere. Correspondence and requests for materials should be addressed to Y.G. (email: yann.guerardel@univ-lille.fr)

First known as a home aquarium fish, zebrafish has been rapidly transformed by scientists into a versatile vertebrate model system to study a wide range of fundamental biological processes in vertebrates, including reproduction, development, evolutionary sciences, genetics, and neurobiology[1–4]. Indeed, the availability of thousands of mutant variants allows in vivo dissection of developmental processes at single cell and molecular resolution. Furthermore, zebrafish exhibits numerous experimental advantages among which are short generation time, high fecundity rate, optic transparency, and fast generation time of functional organs. In addition to be an excellent platform for the study of fundamental biology, zebrafish emerged as a valuable system for modeling human disease[5], including cardiovascular diseases[6,7], cancer onset and progression[8–10], muscular dystrophies[11], skeletal displasias[12], central nervous system disorders[13], and inflammatory bowel disease[14].

About 1–2% of the vertebrate's genome is believed to be involved in the synthesis and processing of glycans that play roles in most, if not all, biological processes. Of uttermost importance, the sialic acids pathway in higher vertebrates requires a large panel of enzymes with various subcellular localizations including the nuclear cytidine monophosphate (CMP)-5-N-acetyl neuraminic acid (Neu5Ac) synthase (CMAS), the cytosolic UDP-N-acetylglucosamine (GlcNAc) 2-epimerase/N-acetylmannosamine kinase (GNE), the cytosolic cytidine monophosphate-N acetylneuraminic acid hydroxylase (CMAH), the Golgi CMP-Neu5Ac transporter (SLC35A1), the Golgi sialyltransferases (ST), and sialidases (Neu)[15]. Thus, it comes as no surprise that any deregulation of glycan synthesis, trafficking, turn-over, or localization may induce variable clinical effects. In this context, zebrafish has recently been used not only to model but also to validate a wide variety of glycan-based diseases related to central nervous system development, cartilage formation, muscular dystrophies, skeletal disorders, neuromuscular transmission defects, neurocutaneous disorder, and inflammatory bowel disease among many others[16–23]. In parallel, the development of innovative methods for imaging glycans in vivo during zebrafish organogenesis provided new insights into the biosynthesis and dynamics of glycans[24–26].

Despite the rush in using zebrafish to study glycosylation-driven Human pathologies, the current lack of a systems wide spatial glycomic view of zebrafish adult tissues and their determinants hinders a more fruitful functional dissection of the underlying molecular mechanisms. So far, only a restricted panel of glycans from the glycomes of zebrafish embryos and oocytes have ever been structurally defined[27–31]. These earlier studies established that zebrafish has the potential of synthesizing not only human-like glyco-epitopes, such as Lewis-x, Galβ1-4(Fucα1-3)GlcNAc, mono and di-sialylated T-antigens, ±Neu5Acα2-3Galβ1-3(Neu5Acα2-6)GalNAc, but also species-specific glycan motifs, such as the Galβ1-4(Neu5Ac/Gcα2-3)Galβ1-4(Fucα1-3) GlcNAc glyco-epitope and Fucα1-3GalNAcβ1-4 (Neu5Ac/Gcα2-3)Galβ1-3GalNAc O-glycans. However, no information on the glycosylation process in the adults was provided. Importantly, zebrafish shows a higher number of paralogous genes compared to Human, as exemplified by the ST (30 in zebrafish vs. 20 in Human)[32,33], due to either species-specific duplications of genes (e.g. ST3Gal1A, ST3Gal1B, ST3gal1C, and ST3Gal1D)[34] or third round genome duplications that occurred in teleosts (e.g. ST6Gal2-r or ST3Gal3-r)[15,34], while some human paralogues are lost (e.g. ST3Gal6)[34].

To provide the requisite structural ground for better-guided correlation of paralogous gene activities and future genetic manipulation in functional analysis of zebrafish glycans, we have initiated a systematic organ-specific profiling of glycoconjugates and the expression patterns of enzymes involved in the terminal glycosylation in adult zebrafish. Among key features, we identified a highest level of sialylation in intestine and ovary that is dominated by 2-keto-3-deoxynononic acid (Kdn) and 5-N-glycolyl neuraminic acid (Neu5Gc), respectively, along with a diverse range of zebrafish-specific terminal glycotopes, O-glycans, and glycosphingolipids (GSLs).

## Results

**Purification of glycans and GSLs.** Eight organs, namely skin, liver, gill, brain, intestines, heart, testis, and ovaries were collected from three separate batches of 30 zebrafishes each and immediately processed according to standard procedure to isolate N-linked glycans (NGs), O-linked glycans (OGs), and GSLs[28,29]. Released glycans were identified and semi-quantified by MS and MS/MS analyses. In parallel, we determined the sialic acid ratios within each of the three families of glycans from different organs, assessed the expression patterns of relevant glyco-genes, and established the tissue-specific expression of selected epitopes. The overall analytical workflow was summarized in Fig. 1.

**Sialic acids composition of organs and glycans.** Three different sialic acid species, Neu5Ac, Neu5Gc, and Kdn, were identified from zebrafish glycoconjugates. As shown in Fig. 2a, the quantity of total sialic acids observed in individual tissues was extremely variable, ranging from 0.6 to 7.8 nmol/mg of proteins for six organs out of eight and up to 22.1 and 41.1 nmol/mg for intestine and ovary. These values are consistent with those reported previously for fish gametes and testis (2–4 nmol/mg proteins), mice tissues (0.4–1.1 nmol/mg), and mouse melanoma cells (14 nmol/mg)[35–37]. Not only the quantity but also the nature of sialic acids exhibited a wide organ-specific variability. The relative quantifications of sialic acids associated with glycans demonstrated similar overall patterns but differed from that observed in total tissues, as shown in Fig. 2b, c. Altogether, these observations confirm that sialic acid synthesis and transfer are differentially regulated in the different organs. The most striking feature of these organ-specific sialylation pattern is the overwhelming presence of Kdn in all glycoconjugates isolated from intestine (from 60% to 85%), with a more modest occurrence in liver, heart, gills, and skin. Another unusual feature that was further explored in the course of present study concerns the modest but relevant content of Neu5Gc in brain glycoconjugates (1–9%). Indeed, brain is considered to be largely devoid of Neu5Gc[38], although its presence has been unambiguously established in the brain glycoconjugates of horse[39].

**Glycomic analysis and cognate sialylation enzymes.** Altogether, we detected 249 NGs, 69 OGs, and 286 GSLs in the eight tissues, some of them being found in several organs while others were organ-specific. We confidently identified 100 different NGs including 95 complex and hybrid NGs (NG1–NG95, Supplementary Data 1), 33 OGs (OG1–OG33, Supplementary Data 2), and 172 GSLs (GL1–GL172, Supplementary Data 3). Out of the 172 GSLs identified, we delineated 52 glycan moieties (G1–G52, Supplementary Data 4) that could each be conjugated to several different ceramide anchors. The structural assignment of all glycans was mostly based on the specific CID MS/MS fragmentation patterns afforded by the $[M + Na]^+$ molecular ions of permethylated glycans in positive mode, as previously described and detailed in Supplementary Notes 1–4 section[28,29,40]. All the identified NGs, OGs, and GSLs were included in Glycome Atlas that is freely available at http://www.rings.t.soka.ac.jp/GlycomeAtlasV5/index.html[41]. Each NG, OG, and GSL was converted to a LinearCode-like format and then incorporated into the Glycome Atlas database, which was augmented to include a

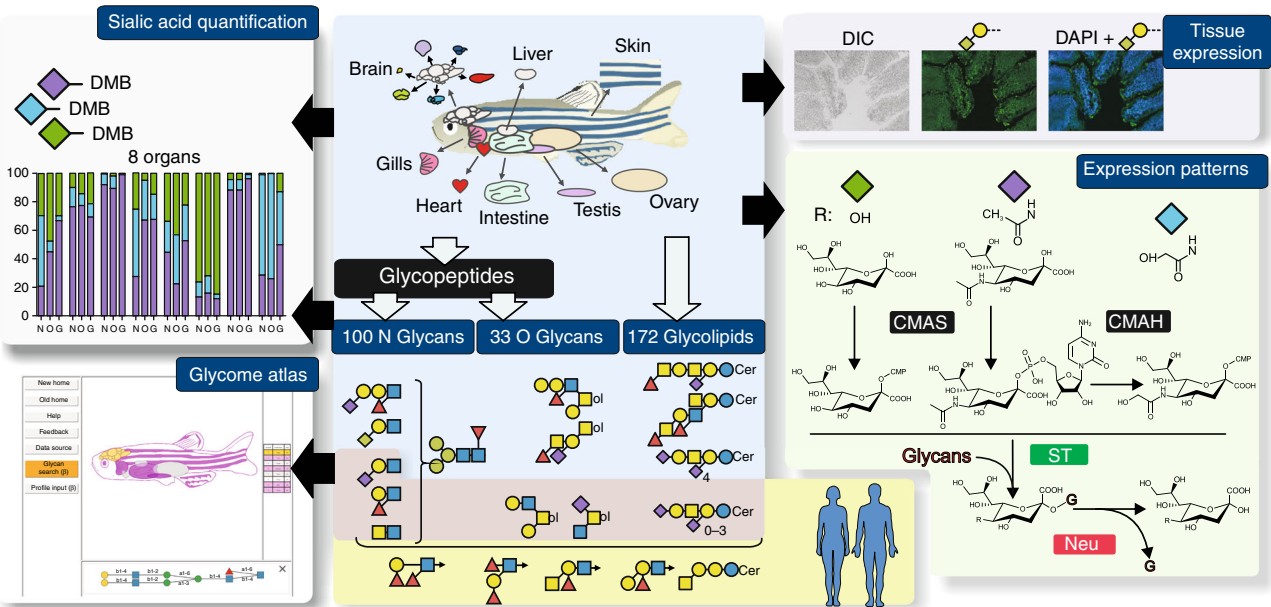

**Fig. 1** Analytical workflow. NGs, OGs, and GSLs were purified from single batches of freshly dissected tissues before analysis of sialic acids, monosaccharides, and oligosaccharide sequences. Unique glycan structures identified and their spatial distribution were posited in an interactive database, Glycome Atlas (http://rings.t.soka.ac.jp/GlycomeAtlasV5/index.html), to provide easy access to the zebrafish glycome. In parallel, the expression pattern of enzymes involved in sialic acid metabolism was established, along with the tissue expression of specific sialylated epitopes. Zebrafish glycome was shown to express many human-type glyco-epitopes, such as type-2 Galβ1-4GlcNAcβ1-based sialyl LacNAc and Leˣ, simple sialylated O-glycans and major gangliosides, but differed by carrying additional glycosyl extensions and the absence of some blood group antigens (A, B), type-1 Galβ1-3GlcNAcβ1-based epitopes, and globo-serie GSLs. CMAS CMP-Sia synthase, CMAH CMP-Sia hydroxylase, ST sialyltransferase, and Neu neuraminidase. Graphical representation is based on accepted conventions for glycans and monosaccharide nomenclature as follows: yellow circle, Gal; yellow square, GalNAc; blue circle, Glc; blue square, GlcNAc; green circle, Man; red triangle, Fuc; purple diamond, Neu5Ac; light blue diamond, Neu5Gc; green diamond, Kdn[77,78]. The interglycosidic bonds between monosaccharides of antennae are represented using the conventional positions as in |, C2 position; /, C3 position; −, C4 position; \, C6 position

user interface for zebrafish. The original LinearCode format did not handle base monosaccharides, such as Hexose or HexNAc, so these were represented as Z and ZN, respectively. This zebrafish data including list of tissues was then added to the Glycome Atlas database. Then the zebrafish image was converted to SVG format so that specific tissues could be clicked to display the corresponding list of NGs, OGs, and GSLs. To use the updated GlycomeAtlas version 5, three buttons are available at the top for human, mouse, and now zebrafish. By clicking on zebrafish, an image of the zebrafish with organs outlined and a table of the eight organs for which glycans have been characterized will be displayed. By clicking on either the image or the table, the selected organ and row in the table will be highlighted in yellow, and the list of glycans profiled will be shown on the right. When a single glycan is clicked in this area, it will be highlighted and its detailed structure will be displayed below the image. Clicking again on another glycan will add the glycan to the selection list below the image. At the same time, the other organs in which the selected glycan(s) are found will also be highlighted in both the image and the table. This allows users to see the tissue specificity of the glycan(s) selected. Right-clicking on a selected glycan will display a pop-up menu for three options: (1) to copy the LinearCode® format for the glycan, which can be used for Glycan Search (available in the menu on the left), (2) open a window to the CFG if the glycan data is available there, or (3) open a window to GlyTouCan, if the glycan data is available[42]. The selected glycans can be deselected by clicking the X on the upper right.

Structural analysis of NGs by combined MS and MS/MS led to identification of 100 NGs out of which 9 were hybrid types and 86 complex types, as described in Supplementary Note 2. The proposed structures and their occurrences were summarized in Supplementary Data 1 and their MS spectra collated in Supplementary Figure 1. The five oligomannosylated NGs assigned to ubiquitous signals at $m/z$ 1580, 1784, 1988, 2192, and 2396 are the so-called Man-5–Man-9 high mannose (Man) type NGs. MS/MS analyses did not reveal any obvious deviation from the canonical structures and thus these five NGs will not be discussed further.

A comparison of the MALDI-MS spectra of NGs isolated from the eight organs established an unexpected level of tissue specificity from both qualitative and quantitative point of views (Supplementary Figure 1 and Data 1). It shows that the proportions of complex NGs versus oligomannosylated NGs are highly variable, ranging from 5% in ovary up to 66% in gills (Supplementary Figure 2). The very low proportion of complex NGs observed in ovary is highly reminiscent of what was observed in embryos[28]. Individual complex and hybrid NGs were relatively quantified by integrating their corresponding MALDI-MS signals from three independent experiments and expressed as % total of complex and hybrid NGs within each sample. The relative abundance of 95 individual complex and hybrid type NGs is summarized as a heat map in Fig. 3 and presented with more details in Supplementary Figure 3. Within each organ, the detected structural heterogeneity varies from 9 distinct NGs in liver up to 46 in gills (Fig. 4a). Only three were identified in all the eight organs: NG89, NG90, and NG91 (Fig. 3, Supplementary Data 1). These are bi-antennary non-core-fucosylated structures carrying the previously defined unique zebrafish epitope Galβ1-4(Neu5Ac/Gc(α2-3)]Galβ1-4(Fucα1-3)GlcNAc[28] on both antennae, sialylated by Neu5Ac and Neu5Gc in 2:0 (NG89), 1:1

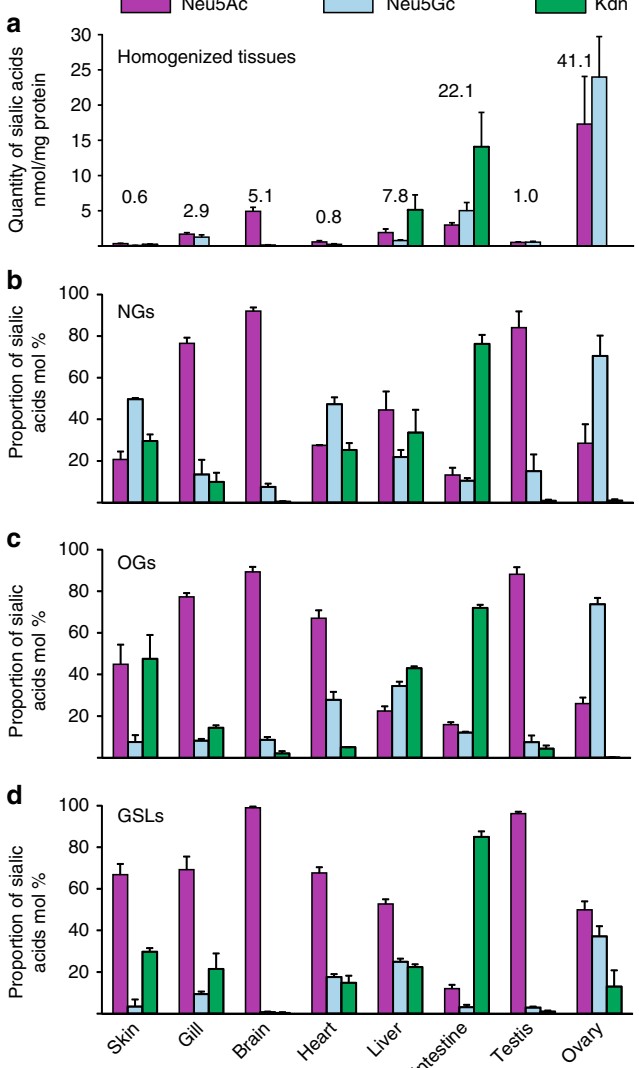

**Fig. 2** Sialic acids composition. Individual sialic acids were identified and quantified by NP-HPLC-FL and LC–MS analysis of (**a**) total tissues, as well as purified (**b**) NGs, (**c**) OGs, and (**d**) GSLs. In total tissues (**a**), quantities were expressed as nmol of sialic acids per mg of proteins. In purified glycans (**b**–**d**), proportions of individual sialic acids were expressed in mol% because the yield of glycan purification from total tissues cannot be precisely established. Purple bar, Neu5Ac; blue bars, Neu5Gc; green bars, Kdn. Data are shown as mean ± SE of at least three experiments

(NG90), and 0:2 (NG91) ratios, respectively. In contrast, the core fucosylated equivalent of NG89 (NG92) could only be identified in low amount in the skin.

Irrespective of the tissue origins, a vast majority of the identified NGs are fucosylated. However, the distribution of fucose between core fucose and Le$^x$-type is more variable. In most tissues, 40–60% of NGs are core-fucosylated and 50–70% carry a Le$^x$ moiety but core fucosylation is notably overrepresented in brain (Fig. 4b). As suggested by the quantification of sialic acids in tissues and NGs (Fig. 2a, b), sialylation patterns of NGs also show important tissue variability both in terms of the degree and chemical nature of sialylation (Fig. 4c). It should be noted that the proportions of sialylated glycans as determined from MALDI-MS signal intensities do not always match the quantification data of sialic acids (Fig. 4c vs. Fig. 2a) because (1) the latter does not take into account individual intensities, (2) the nature of sialic acids affects the detected signal intensities of glycans in MALDI-MS,

and (3) MS analysis would bias against and thus underestimate the larger, potentially highly sialylated glycans with less favorable ionization, and detector response factor.

The vast majority of identified sialic acids, irrespective of their nature, was attached via α2-3 linkage to internal Gal residue of the ±Galβ1-4(Neu5Ac/Gcα2-3)Galβ1-4(±Fucα1-3)GlcNAc motif. This unique sialylated motif was defined by MS/MS analysis and confirmed by GC/MS linkage analysis of total NG fraction that demonstrated the presence of C3-substituted, C4-substituted, and C3, C4-disubstituted Gal residues, whereas C6-substituted Gal residues could not be detected. However, we cannot preclude the possibility that minor Siaα2-6Gal-containing N-Glycans may exist since our MS/MS data could not discriminate α2-3- from α2-6-sialylation on galactose (Gal). Hanzawa and collaborators have previously demonstrated that α2-6-substituted N-glycans were the predominantly expressed sialylated N-glycans in deyolked embryos[31]. In accordance with their study, we have detected α2-6 sialylation on the N-acetylgalactosamine (GalNAc) residue of LacdiNAc motifs in NG15, 24, 58, 59 that were exclusively observed in ovary in our study. However, they also showed that (1) adult-derived yolk does not contain α2-6-linked sialic acid, and (2) α2-6 sialylation decreases during embryonic development from 6 to 48 hpf, whereas α2-3 sialylation increases in the later developmental stages. These observations are consistent with our current data and collectively suggest a developmental shift from embryonic α2-6 to adulthood α2-3-sialylation along with an overall significantly down-regulated sialylation to either absent or very minor level in all adult organs (NG20 and NG60). Contrary to Neu5Ac and Neu5Gc, Kdn was only found on the Gal residue of Gal(β1,4)GlcNAc but never detected on the additionally galactosylated or fucosylated terminal disaccharide unit. At present, the sialyl linkage of Kdn to Gal could not be unambiguously established due to the low amount of Kdn-substituted glycans.

Altogether, our structural analyses established that the terminal glycosylation units decorating the zebrafish N-glycome are almost entirely based on the type 2 LacNAc (Galβ1-4GlcNAc) antennae differentially substituted by Fucα1-3 and Siaα2-3 on the GlcNAc and Gal residues, respectively, to yield the Le$^x$ and SLe$^x$ epitopes commonly expressed on mammalian glycoproteins. However, it can also be further substituted by a terminal Galβ1-4 residue that has never been observed in Human and other mammals, to the best of our knowledge. In addition, a number of GalNAcβ1-3/4GlcNAc antennae that can be further sialylated on the terminal GalNAc residues were identified in several organs, with a clear prevalence in ovary, testis, and skin. Although the β1-4 version of this unit, also called LacdiNac, is abundant in invertebrates[43,44], it is less commonly found in mammalians species including Humans[45,46], where it is considered as a potential diagnosis marker for a number of cancers[47,48].

Thirty-three different mucin type OGs from eight organs could be identified by MS (Supplementary Figure 4) and MS/MS analyses of permethylated derivatives, as described in Supplementary Note 3. Their structures and relative quantifications are summarized in Fig. 5, Supplementary Figure 5, and Supplementary Data 2. These analyses did not lead to identification of other types of O-glycans, such as O-Man, O-Fuc, or O-Glc that may however be present at lower quantities. The major O-glycans are based on core-1 Gal(β1-3)GalNAc-ol and core-2 Galβ1-3(GlcNAcβ1-6) GalNAc-ol structures that are widely distributed among mammalians including Humans. Few core-3 GlcNAcβ1-3GalNAc-ol and core-4 GlcNAcβ1-3(GlcNAcβ1-6)GalNAc-ol-based structures were additionally detected as very minor components. Interestingly, zebrafish exhibited some core extension and terminal glycosylation specificities never observed in Humans including those based on the GalNAcβ1-4Galβ1-3GalNAc-ol sequence that

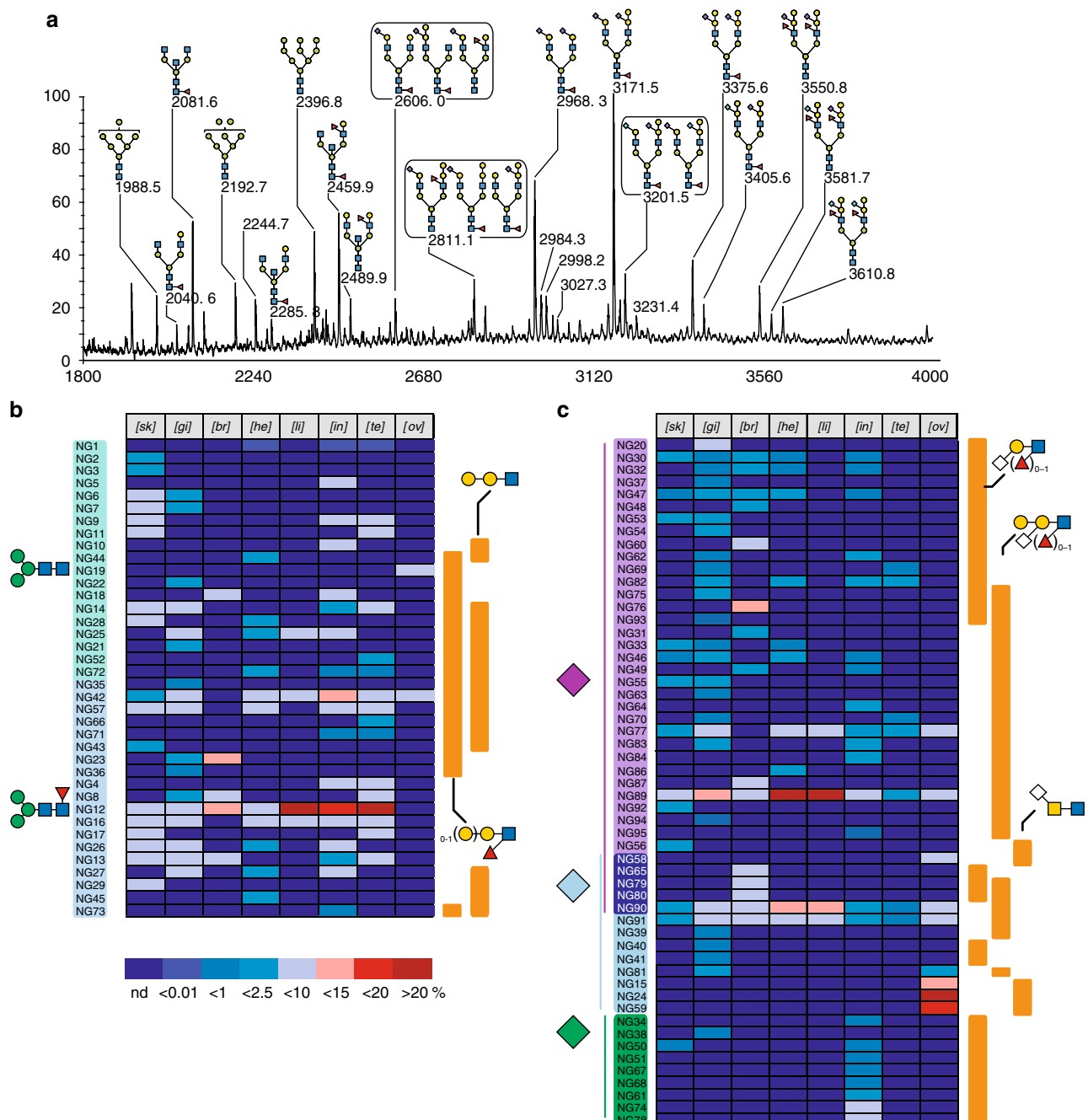

**Fig. 3** *N*-glycome profiles. The chemical nature of individual glycans was established by a combination of MS/MS, GC/MS, and NMR, in comparison with previous work[28,29] and reported in Supplementary Figure 1 and Supplementary Data 1. **a** Representative MALDI-TOF MS spectra of permethylated NGs isolated from brain. All MS spectra are provided in Supplementary Figure 1. **b** Tissues-specific distribution and relative quantification of neutral and (**c**) sialylated (right panel) NGs according to following glycan epitopes: core fucosylation, (Galβ1-4)$_{0-1}$Galβ1-4(Fucα1-3)GlcNAcβ, Galβ1-4Galβ1-4GlcNAcβ for neutral and nature of substituting sialic acids (Neu5Ac, Neu5Gc, Kdn), Siaα2-3Galβ1-4(Fucα1-3)$_{0-1}$GlcNAcβ, Galβ1-4(Siaα2-3)Galβ1,4(Fucα1-3)$_{0-1}$GlcNAcβ, and Siaα2-6GalNAcβ1,3/4GalNAcβ. All Sia residues (Neu5Ac, Neu5Gc, Kdn) linked to Gal were considered as α2,3 linked although minor α2,6-Gal may co-exist as described in the text. Relative quantifications of individual compounds were extracted from the dataset presented in Supplementary Figure 3

was very abundant in testis and ovary, and the Galβ1-4Galβ1-4 (Fucα1-3)GlcNAc sequence that is reminiscent of the one observed in *N*-glycans.

As for NGs, OGs exhibited a high degree of tissue specificity (Fig. 5, Supplementary Figure 5, Supplementary Data 2). The level of fucosylation varied from 20% in brain to 80% in ovary, which suggests a tight organ-specific regulation of the fucosylation process. In accordance with the total sialic acid quantification,

Neu5Ac appears to be the most widely distributed sialic acid among OGs in all tissues except ovary in which Neu5Gc predominates (Figs. 2c, 5b, c). Kdn-sialylated glycans could only be firmly identified by MS and MS/MS analyses in intestine as five core-1 OGs (OG1, OG10, OG11, OG12, and OG13, Fig. 5b). Despite the relatively lower heterogeneity of O-glycosylation compared to *N*-glycosylation, we could not identify any glycan that is common to all organs. The most widely distributed OGs

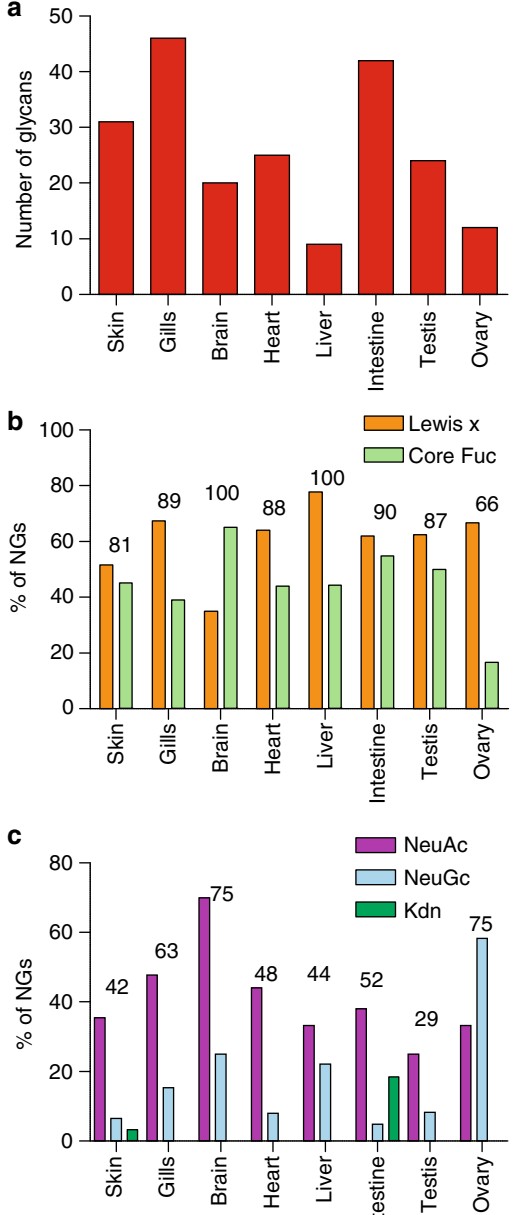

**Fig. 4** Comparison of NGs features between organs. **a** Number of NGs confidently identified in each organ. **b** Fucosylation pattern expressed as the proportion of NGs substituted by one or more fucoses on the chitobiose disaccharides (green bars, core Fuc), and/or in Lewis-x epitope (orange bars); numbers indicate the overall percentage of fucosylated NGs. **c** Sialylation pattern expressed as the percentage of NGs substituted by any of the three sialic acids (purple bars, Neu5Ac; light blue bars, Neu5Gc; green bars, Kdn); numbers indicate the overall percentage of sialylated NGs relative to total NGs in the corresponding organs. It should be noted that a single NG may bear both Lewis-x and core Fuc, as well as different types of sialic acids

are Galβ1-3(Neu5Acα2-6)GalNAc-ol (OG2), and Neu5Acα2-3Galβ1-3(Neu5Acα2-6)GalNAc-ol (OG15) in seven organs out of eight. Adult organs exhibited an overall much higher structural diversity than early embryos that are dominated by core 1 O-glycans OG18 and 20. It is noteworthy that the O-glycome profile of ovary is very similar to that of 1 h post-fertilized embryos. Other than these major glycans, two very minor Lex-extended core 2 O-glycans and the oligosialylated versions of

NeuGc-substituted OG20 were identified in embryos but not observed in adult tissues[28]. These minor O-glycans are believed to be mostly associated with soluble glycoconjugates in the perivitelline space, but not embryonic tissue[27]. Their disappearance in adult organs is in agreement with decreasing oligosialylation of glycoproteins along zebrafish embryonic development[27].

As for N-glycans and O-glycans, glycolipids were extracted, purified, and permethylated prior to MS analysis (Fig. 6, Supplementary Figure 6, Supplementary Note 4). In addition, monosaccharide and lipid composition were confirmed by GC/MS. The structural analysis data are summarized in Supplementary Data 3 and 4. In all samples, we observed a high proportion of Hex-Ceramide (Hex-Cer) and lacto-Ceramide (Lac-Cer) that represent about 80% of total GSLs. Alongside Hex-Cer and Lac-Cer, we identified a complex pattern of GSLs dominated by gangliosides (Gg) and lacto-ganglioside (LcGg) series. Based on known biosynthetic pathways of the glycan moieties of GSLs in vertebrates and MS/MS analysis, we assigned 46 GSLs to Gala series (Galβ1-1′Cer), 20 to Hemato series (Galβ1-4Glcβ1-1′Cer), 1 to Lacto/Neolacto series (Galβ1-3/4GlcNAcβ1-4Galβ1-4Glcβ1-1′Cer), 49 to Lactoganglio series (GlcNAcβ1-3(GalNAcβ1-4)Galβ1-4Glcβ1-1′Cer), and 56 to Ganglio series (GalNAcβ1-4Galβ1-4Glcβ1-1′Cer) (Supplementary Data 3). The 172 GSLs (GL1-172) were made of 52 different carbohydrate moieties (G1-52) substituted by multiple ceramides, as summarized in Supplementary Data 4.

As revealed by MS mapping, GSLs exhibited a very high degree of organ-specificity (Fig. 6b, Supplementary Data 3 and 4) contributed by both glycan and ceramide moieties. The relative quantification of sphingoid bases revealed that d18:1 sphingosine was the major component for all organs (Fig. 6c). Additionally, significant proportions of t18:0 phytosphinganine were observed in skin, liver, intestine, testis, and ovary, whereas d18:0 sphinganine was identified in skin and testis. MS/MS analyses showed that the d18:1/C16:0 and d18:1/C24:0 combinations were the prevalent forms of ceramides, except for brain in which GSLs were almost exclusively substituted by the unusual d18:1/C18:0. Glycans moieties also exhibited a high level of organ-specificity with respect to the proportions of different GSLs families, level of fucosylation, and sialic acids diversity. As observed for NGs and OGs, the number of GSLs varies from one organ to the other (from 27 GSLs in liver to 43 in ovary) and their distribution in structurally related families is totally different (Fig. 6c). In particular, ovary stands out by the prevalence of the LactoGanglio series (>60%), whereas this family of GSLs is absent in brain and heart and below 20% in remaining organs. It is noteworthy that the fucosylation level of GSLs in different organs was more variable than in OGs and NGs (Fig. 6c). Indeed, the proportion of fucosylated GSLs varied from 0% in brain to about 40% in ovaries. Finally, the distribution of sialic acids in each organ was comparable to OGs and NGs, showing a prevalence of Neu5Ac-sialylated GSLs in all organs (Fig. 6c). The only discrepancy is the ovary's GSLs that were predominantly substituted by Neu5Ac whereas OGs and NGs were predominantly substituted by Neu5Gc. Finally, as observed for NGs and OGs, intestine was the only organ in which Kdn sialylated a significant proportion of GSLs.

Considering the prevalence of sialic acids in zebrafish glycome, we paid a particular attention to the sialylation machinery. As a first step, we screened databases for the in silico identification of Danio rerio ST genes, sialidases (neu), CMP-NeuAc synthase (cmas), and the CMP-NeuAc hydroxylase (cmah)[33,49]. We identified 11 α2,6-ST (st6gal and st6galnac), 10 α2,3-ST (st3gal), 7 α2,8-ST (st8sia), 2 CMAS (cmas1 and cmas2), 7 sialidases (1 neu1, 5 neu3 and 1 neu4), and 1 cmah, and oligonucleotide primer pairs were synthetized as described previously for each of

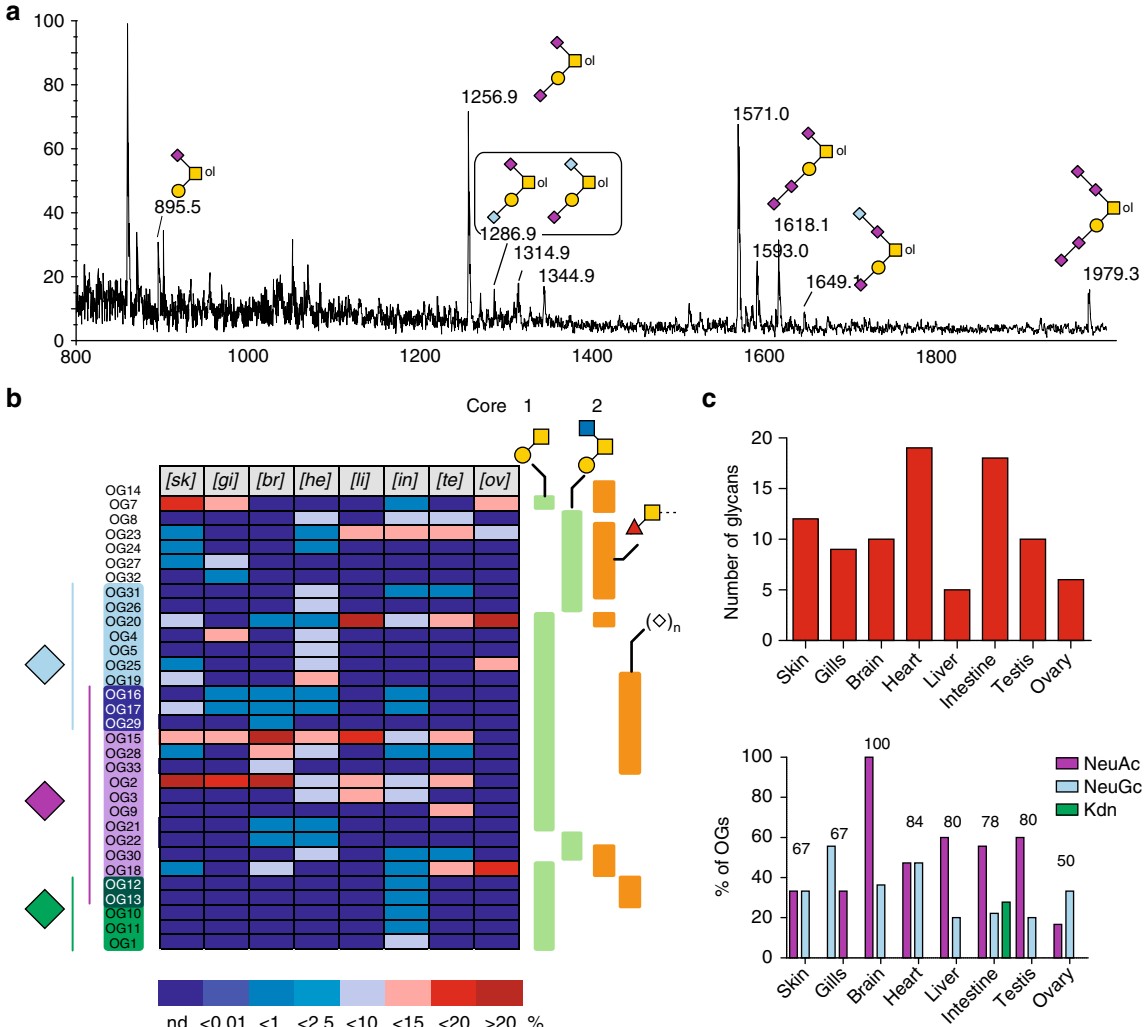

**Fig. 5** *O*-glycome profiles. **a** Representative MALDI-TOF MS spectra of permethylated OGs isolated from brain. All MS spectra are provided in Supplementary Figure 4. **b** Tissues-specific distribution and relative quantification of OGs according to following glycan epitopes: nature of substituting sialic acids (Neu5Ac, Neu5Gc, Kdn), core-1 Galβ1-3GalNAc, Core-2 Galβ1-3(GlcNAcβ1-6)GalNAc, Fucα1-3GalNAcβ, and oligosialylation. The nature of individual glycans was established by a combination of MS/MS, GC/MS, and NMR, in comparison with previous work[28,29]. Tissue-specific distribution of glycans is presented in Supplementary Data 2. Relative quantifications of individual compounds were extracted from dataset presented in Supplementary Figure 5. **c** Number of OGs identified in each organ (upper panel) and proportion of sialylated OGs in each organ expressed as the percentage of OGs substituted by any of the three sialic acids (purple bars, Neu5Ac; light blue bars, Neu5Gc; green bars, Kdn); numbers indicate the overall percentage of sialylated OGs relative to total OGs in the corresponding organs. It should be noted that a single OG may bear different types of sialic acids

the ST[29], sialidases[50], CMAS, and CMAH[51] (Supplementary Data 5). RT-PCR was used to establish the profile of expression of these genes in nine adult zebrafish tissues, i.e. ovary, testis, higher intestine, lower intestine, brain, heart, gill, skin, and liver (Fig. 7 and Supplementary Figures 7, 8). The recently described α2,3-ST *st3gal1*-related genes (*st3gal1C* and *st3gal1D*), *st3gal3*[34] and the membrane-associated sialidase *neu4*[50] were not expressed at detectable level, whereas the sialidase genes *neu3.5* and *neu1*[50], the teleost-specific α2,3-sialyltransferase gene *st3gal3-r*[34], and the 2 synthases *cmas1* and *cmah* were expressed in all examined tissues. The *st3gal5* gene also known as the GM3 synthase is expressed in all tissues examined, which correlates well with the wide expression of GM3 and GM2 Gg. In addition, the *st8sia1* and *st8sia5* genes known as GD3 and GT3 synthases (SAT-II and SAT-V, respectively) shows a profile of expression restricted to brain, in agreement with diSia/oligoSia-containing GSLs observed in brain (Fig. 6b). Mono-α2,3-sialylated core 1 structures (OGs) found in heart likely arise as a consequence of ST3Gal II activity since none of the *st3gal1*-related genes were expressed in heart. In

intestine, the monosialylated and disialylated OGs structures observed likely result from the enzymatic activity of ST6GalNAc IA, ST6GalNAc IB, ST3Gal IA since high expression of the corresponding genes was observed in this tissue. In addition, diSia epitopes on OGs were also identified in intestine, brain, and heart that probably resulted from ST8Sia VI enzymatic activity previously described for the mouse and human enzymes[52,53].

**Intestine sialome**. Among the organs that were screened, some exhibited remarkable or unexpected features. One of the most surprising features was the marked prevalence of Kdn for intestine (Fig. 2). Accordingly, monoclonal antibody raised against Kdn(α2,3)Gal epitope (KDN3G) showed a strong response for two high molecular weight broad bands around 125 kDa and above 250 kDa in intestine, as well as in liver and heart although of lower intensities (Fig. 8a). Intestine also contained mucin-like components too large to reach separating gels (Supplementary Figure 9). Immunochemical detection of

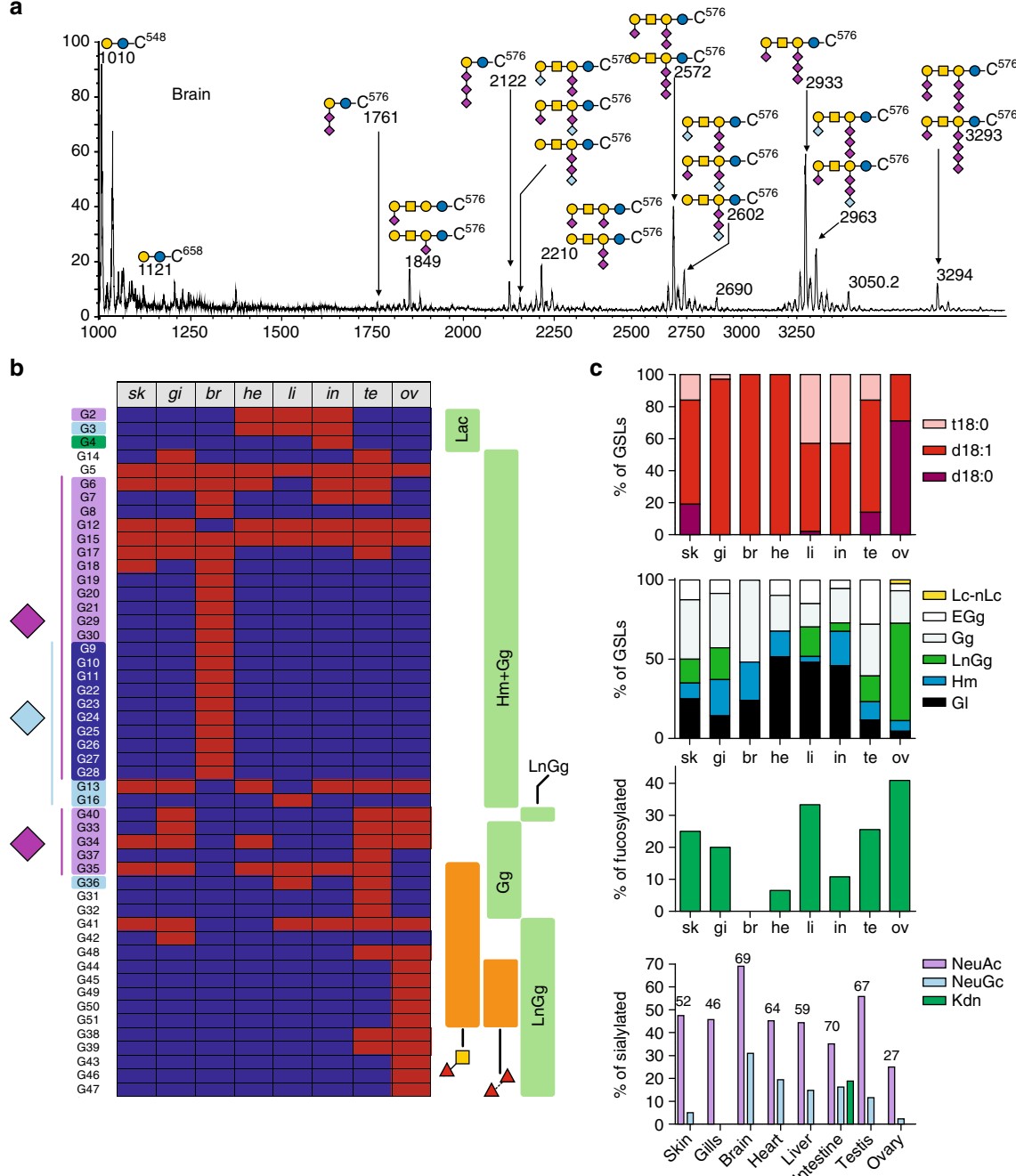

**Fig. 6** Glyco-lipidome profiles. **a** Representative MALDI-TOF MS spectra of permethylated GSLs isolated from brain. All MS spectra are provided in Supplementary Figure 6. **b** Tissue-specific distribution of the glycan moieties of GSLs according to following glycan epitopes: nature of substituting sialic acids (Neu5Ac, Neu5Gc, Kdn), the GSL families (Lacto; hemato and ganglio Hm + Gg; extended galglio ExGg; lacto-ganglio LnGg), Fucα1,3GalNAcβ, and oligo-fucosylation. The presence of multiple ceramides prevents reliable quantification. The identified species are depicted in red and the non-detected ones in dark-blue. **c** From top to bottom panels: distribution of ceramides expressed as the percentage of GSLs bearing one or the other sphingoïd base (dark red bars, d18:0; medium red bars, d18:1; light red bars, t18:0); distribution of families expressed as the percentage of GSLs from one or the other family; occurrence of fucosylation expressed as the proportion of GSLs substituted by at least one fucose residue; sialylation pattern expressed as the percentage of GSLs substituted by any of the three sialic acids (purple bar, Neu5Ac; light blue bars, Neu5Gc; green bars, Kdn); numbers indicate the overall percentage of sialylated GSLs relative to total GSLs in the corresponding organs. It should be noted that a single GSL may harbor different types of sialic acids

Kdn using KDN3G antibody showed a specific signal at the surface of epithelial cells lining the lumen of zebrafish intestine (Fig. 8b). They appear as intense dot-like particles, in agreement with their putative attribution to aggregates of secreted mucins. However, it should be noted that the presence of Kdn is not restricted to OGs, so KDN3G may also target NG-substituted glycoproteins and membrane glycolipids. Compared to intestinal epithelial cells, hepatocytes cells surface exhibited evenly distributed KDN3G-reactive epitopes, suggesting that Kdn was not carried on mucin-like secreted glycoproteins, in accordance with WB analysis (Supplementary Figure 9b).

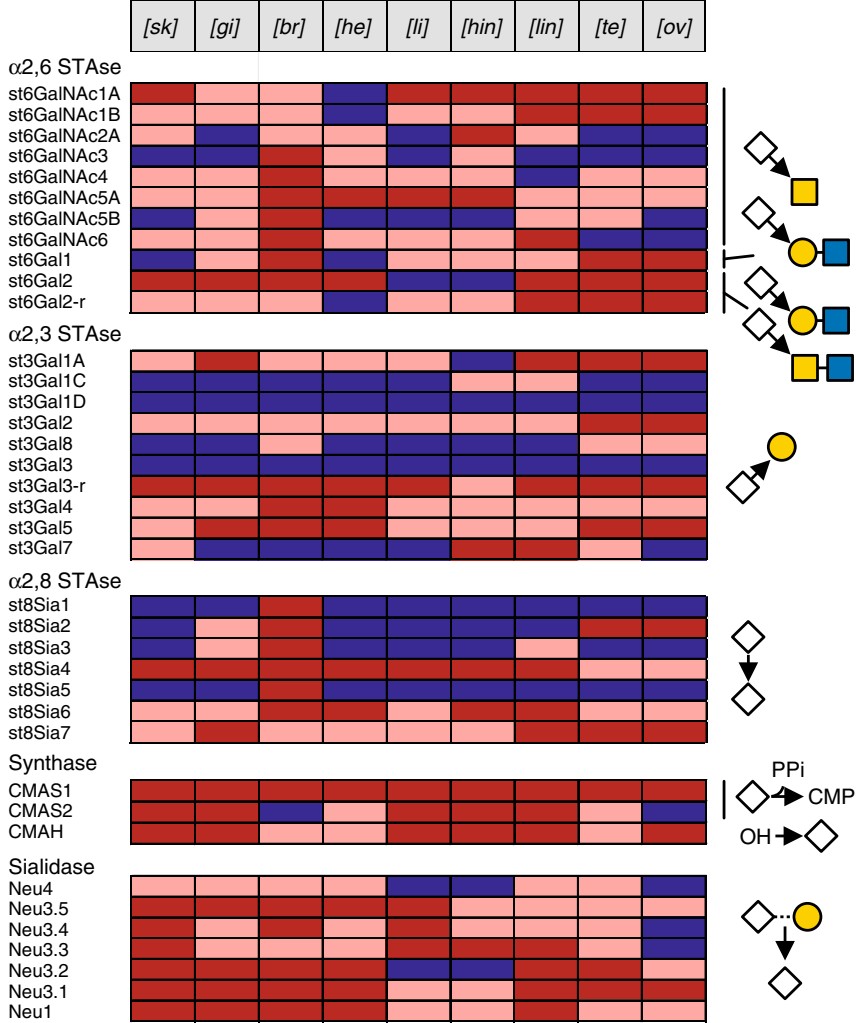

**Fig. 7** Sialo-genes expression. Expression profiles of sialyltransferases, CMP-sialic acid synthases (CMAS), sialidases (Neu), CMP-sialic acid hydroxylases (CMAH), and sialidases genes in various zebrafish tissues based on RT-PCR data (Supplementary Figure 7). Expression levels of single mRNAs were depicted according to their relative intensities in each organ as non-detected (blue), intermediate intensity (pink), and strong intensity (red). Putative activities based on phylogenetic analyses are depicted at the right side of the map

Altogether, mass spectrometry, sialic acids quantification, Western-blot, and immunochemistry analyses established that intestine is the only organ that contains significant amount of Kdn-substituted glycans. In order to provide clues about the molecular bases that drive such exquisite organ specificity we tried to identify Kdn in the input and output of zebrafish. Whereas sialic acid analysis did not detect appreciable amount of Kdn in the fish food, it demonstrated that the intestine lumen contained significant amount of sialic acids (Neu5Ac 3 pmol/mg, Neu5Gc 5 pmol/mg, and Kdn 37 pmol/mg of dry material). Although variable from fish to fish, similar proportions of sialic acid species were recovered in the excreted feces. Feces components were separated by Nycodenz® density gradient using a recently developed procedure[54]. Composition analysis showed that Kdn was highly enriched in purified microbiota, but not in the insoluble food debris (Fig. 8c). Furthermore, a vast majority of Kdn appeared to be associated with glycoconjugates, presumably bacterial polysaccharides, rather than present as free form. This strongly suggests that the microbiota harbored in the fish intestine independently synthesizes large amounts of Kdn. This view is reinforced by the discovery that the intestinal symbiont *Bacteroides thetaiotaomicron* can synthesize Kdn and transfer it to its capsular polysaccharides[55]. However, considering the structure of Kdn-substituted glycans, including

those established for OGs, NGs, and GSLs, the identified glycans undoubtedly originate from zebrafish and not microbiota. Thus, it is conceivable that the high content of Kdn in intestine results from the direct scavenging of Kdn present in the lumen by the epithelial cells to build up zebrafish glycans in an energy saving manner (Fig. 8d). The scavenging of sialic acids through the intestinal barrier has been amply documented and could at least partially explain the clear tropism of Kdn for the intestinal tissue[56]. However, it is hypothesized here that it may directly influence the intestinal glycosylation pattern through its recycling to epithelial mucin-type glycoconjugates, among other components. Alternatively, Kdn might also be directly transferred to extracellular glycoconjugates by exogenous bacterial sialyltransferase activities, but the structural similarity of Neu5Ac and Kdn-substituted glycans is clearly in favor of an endogenous sialyltransferase activity. Scavenged Kdn may also be secondarily transported to other organs through blood circulation particularly to liver, to be used in glycoconjugates metabolism. A complementary line of explanation may arise from the expression of two CMP-sialic acid synthases (CMAS) in zebrafish. Indeed, in contrast to other vertebrates, the genome of teleost fishes from the Otocephala clade contains two homologs of *Cmas* genes, dre*Cmas1* and dre*Cmas2* in zebrafish[51]. In zebrafish, both genes seem to be expressed as functional enzymes

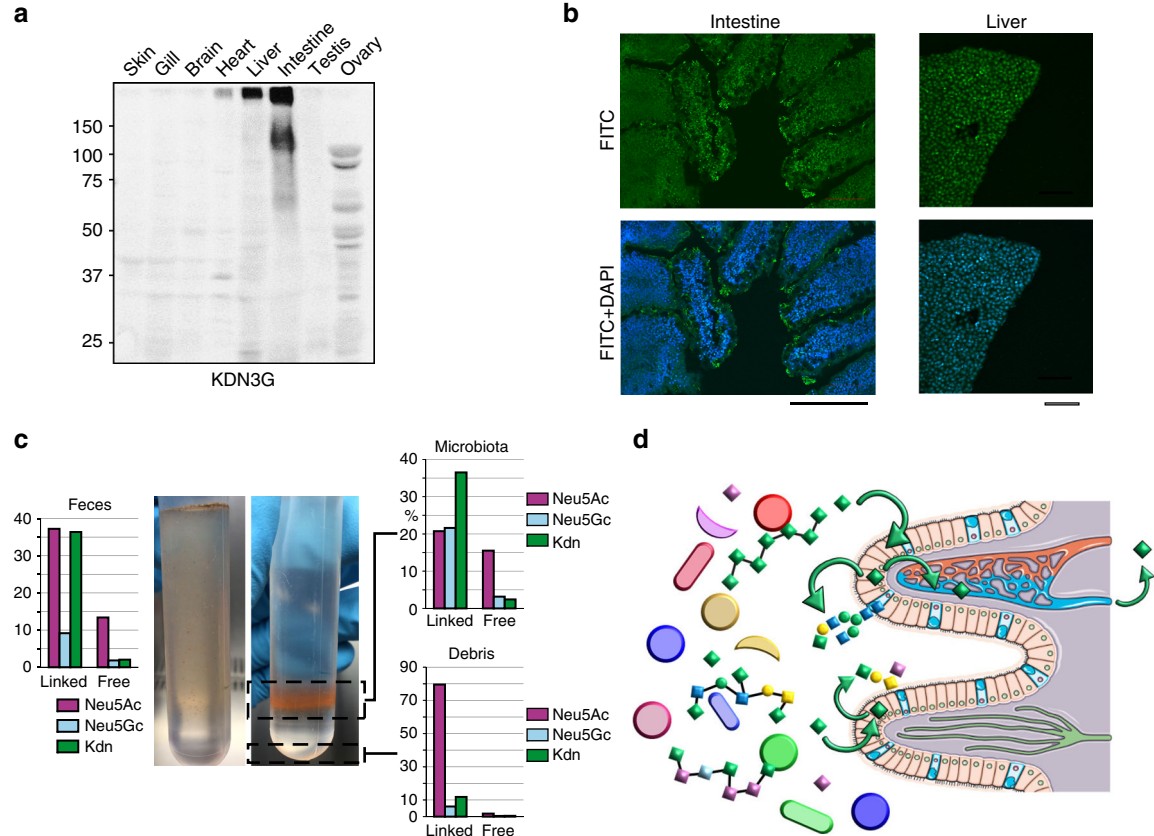

**Fig. 8** Distribution of Kdn in organs. **a** WB screening of organs using Kdn(α2,3)Gal specific-epitope KDN3G antibody raised against GM3-Kdn and characterized to recognize a purified KDN-containing glycoprotein (Gp-Kdn) derived from rainbow trout ovarian fluid[11,76]. Control experiments using IgM/G are shown in Supplementary Figure 9b. **b** Histochemistry staining using DAPI and KDN3G showed the presence of Kdn(α2,3)Gal at the surface of intestine epithelium and liver cells. Scale bar corresponds to 100 μm for intestine and 50 μm for liver. Control experiments using IgG are shown in Supplementary Figure 10. **c** Sialic acid composition analysis of feces before (left panel) and after (right panel) purification of by Nycodenz® gradient centrifugation shows a preferential association of Kdn with glycoconjugates of microbiota. **d** Hypothesized recycling of bacteria associated Kdn (green diamonds) by epithelial cells into endogenous glycoconjugates and subsequent transfer into blood circulation

that differ not only by their subcellular localization but also by their substrate specificity. Whereas dreCmas1 exhibits highest activity toward Neu5Ac, dreCmas2 preferentially activates Kdn[51]. Based on this finding, we have assessed the presence of the two paralogous genes in the different tissues by RT-PCR. As shown in Fig. 7 and Supplementary Figure 7, whereas dreCmas1 was ubiquitously expressed at similar levels in all organs, dreCmas2 exhibited a more restricted expression pattern. In particular, it was barely observed in ovary, brain, and heart but showed a robust expression in intestine, liver, and skin, as well as an intermediate expression in gills and testis. Interestingly, organs that showed the highest expression levels are precisely the ones that seem to contain the highest levels of Kdn associated with their glycoconjugates. This observation does not provide a single explanation for the tropism of Kdn toward a restricted set of organs, but one may speculate that the availability of Kdn substrate have driven the expression of a Kdn-specific Cmas and that the two observations are connected to the differential synthesis of Kdn sialoconjugates. Interestingly, the recently described st3gal5-related gene renamed st3gal7[34], which corresponds to the GM4 synthase[57], shows a profile of expression rather similar to dreCmas2 suggesting that st3gal7 may also exhibit a certain specificity toward the transfer of Kdn on Gal-Cer moiety.

**Brain sialome**. The glycome of brain also stands out owing to its unique sialylation pattern. First, brain is the only organ in which glycans harbor substantial amount of oligosialylated motifs. In

particular, about 90% of the glycan moieties of GSLs from brain are sialylated and 75% are substituted by Sia2 to Sia4 epitopes. The presence of abundant oligosialylation in brain GSLs was confirmed by HPLC analysis of DMB-derivatized oligosialyl motifs, as shown in Supplementary Figure 11. Oligosialylation extends either from the Lac-Cer core of the hemato series (GL60-66; G7-11) or from the internal Gal residue of ganglio series trisaccharide core (GL95-106; G19-30) up to the newly described GP1d (GL106; G30). Gg are well known to represent the major type of GSLs in the brain of all inspected vertebrates so far. However, whereas the GSLs of mammal brains, including humans, are dominated by mono-sialylated, di-sialylated, and tri-sialylated GSLs GM1, GD1a, GD1b, and GT1b, zebrafish brain is dominated by the tri-sialylated, tetra-sialylated and penta-sialylated GSLs GT1b, GT1c, GQ1c, and GP1d. The prevalence of higher form of sialylation in fish GSLs has already been documented in the brain of several fish species including codfish, skate, and dogfish[58–60]. Similar to GSLs, all the OGs isolated from brains are sialylated, harboring up to four sialic acids. Brain synthesizes a very limited panel of OGs made of differentially sialylated Gal(β1-3)GalNAc-ol core-1 molecules sialylated on the C6 position of GalNAc-ol by either Neu5Ac or Neu5Gc and potentially substituted on the C3 position of Gal residue by one or two sialic acids or on the C8 position of Sia(α2-6)GalNAc-ol. Accordingly, brain expresses the most diverse panel of ST (Fig. 7 and Supplementary Figures 7, 8) being the only organ that expresses all eight members of st6galnac genes and three members of the st6gal

genes, which would explain the prevalence of sialylated OGs in brain. Similarly, the restricted expression of ST8Sia1 and ST8Sia3, which are believed to be involved in the elongation of sialylated chain of GD3 and GT3, is in agreement with the specific synthesis of b-series, c-series, and d-series of GSLs in brain. Finally, although the analytical techniques did not allow direct observation of polysialylated epitopes, both st8sia2 and st8sia4 genes responsible for the polysialylation of neural cell adhesion molecules (N-CAM), were expressed in the zebrafish brain as described previously in the rainbow trout brain[61] and in accordance with previous immunochemistry study[62].

As mentioned before, Neu5Gc is known to be expressed in minute amounts in the brain of vertebrates and its presence in neural tissues is still open to debates[38]. Here, as many before us, we established that brain was the organ that contains the lowest amount of Neu5Gc, ranging from 1% of total sialic acids in GSLs to 8% in OGs[63]. The low level of Neu5Gc synthesis in brain was substantiated by the relative low expression of CMP-Neu5Ac hydroxylase (CMAH) compared to other tissues (Fig. 7 and Supplementary Figure 7) as previously observed[64]. Although low, the amount of Neu5Gc is not negligible and its presence in individual glycans was consistently proven by MS/MS analysis. Of particular relevance, the presence of three major NGs bearing the so-called zebrafish epitopes Galβ1-4(NeuAc/Gcα2-3)Galβ1-4(Fucα1-3)GlcNAc (NG89-91) and several Neu5Gc-substituted Gg (GL98-99, GL101, GL103-104) was firmly established. As shown in Supplementary Figure 12, the comparison of the MS/MS fragmentation patterns of GT1 molecules (GL96 and GL97 versus GL98, GL99, GL100, and GL101) and GP1 molecules (GL102 versus GL103 and 104) unambiguously established that Neu5Gc can replace Neu5Ac at multiple positions. The presence of Neu5Gc-substituted Gg in brain has already been firmly established in different species, which leaves no doubt concerning the capacity of brain tissue to process Neu5Gc[39,65,66]. Although structural analysis of total brain cannot unambiguously answer to the nagging question about the localization of Neu5Gc-substituted glycans, the data provided here supports the localization of Neu5Gc in neural tissues rather than in endothelial tissues or blood cells in two ways. First, the glycan moieties of the Neu5Gc-substituted glycans identified in zebrafish brain are identical to those of the Neu5Ac-substituted glycans. This is of particular relevance for GSLs, because oligosialylated Gg are known to be very specific of neural tissues. As shown in Fig. 6a and Supplementary Data 3, the pattern of Neu5Gc-GSLs is identical to Neu5Ac-GSLs. Second, not only the glycan moieties of Neu5Ac and Neu5Gc containing GSLs are identical, but also their lipid moieties. Indeed, Neu5Gc containing GSLs are made of the specific d18:1/C18:0 ceramide ($m/z$ 576) that was exclusively observed in brain GSLs. Thus, similarities between Neu5Gc and Neu5Ac containing glycans strongly suggest that neural tissue synthesizes small but significant amounts of Neu5Gc containing glycoconjugates.

Finally, considering the specificities of brain glycomes, we investigated if some of these features were shared among the different regions of brain. Toward this aim, we have separated the zebrafish brain into six defined regions as previously described[67], including olfactory bulb, telencephalon, optic tectum, cerebellum, medulla, and hypothalamus. Considering the minute amount of tissues at disposition, we focused on the NGs. In agreement with previous observations, the Neu5Gc proportion in individual regions was lower than 1%. Analyses of Neu5Gc content using nine replicates did not show significant differences between regions. Similarly, MS analysis of N-glycomes did not reveal major differences between regions (Supplementary Figure 13). Indeed, most of the major NGs identified in whole brain could be observed in individual regions albeit in different proportions. As

for whole brain, most regions showed a high proportion of complex NGs except cerebellum, which showed a higher proportion of oligomannosylated NGs than complex NGs. MS/MS analysis of major complex NGs confirmed the prevalence of core fucosylated NGs, as was observed in whole brain. In particular, NG76 that appears as the major brain-specific NG was observed in all regions, albeit also in different proportions. The presence of Neu5Gc-substituted NGs could be established by MS and MS/MS in some region, in particular in olfactory bulb (Supplementary Figure 13). Among those, the ubiquitous Neu5Gc containing NG90 and NG91 could be clearly identified exclusively in the olfactory bulb from three different independent preparations.

## Discussion

In conclusion, the present study established that zebrafish synthetizes a rich and diverse glycome that share numerous features with higher vertebrates including the expression of fucosylated and sialylated terminal LacNAc and LacdiNac epitopes, simple sialylated O-glycans and Gg expressions. Zebrafish glycans are decorated with species-specific and organ-specific glycans epitopes that arise from unique extensions of mammalian structural motifs, notably the Gal(β1,4)-extended version of LacNAc unit and GalNAc(β1,4)-extended core-1 OGs. On the other hand, zebrafish lacks common mammalian epitopes, such as those based on type-1 chain (Galβ1-3GlcNAc, Le[a], Le[b], etc.) and major GSLs families like globosides. More importantly, the present report demonstrated that systems glycomics could uncover so far untapped physiological processes, such as scavenging and metabolic incorporation of bacterial monosaccharides in intestine. Altogether, we believe that the glycosylation map of zebrafish, in conjunction with the expression pattern of specific glycoenzymes, provides a unique opportunity for the zebrafish biology and glycobiology communities to address the glycans functions on a solid structural background.

## Methods

**Zebrafish handling**. Wild-type *Danio rerio* AB stock was purchased from the Zebrafish International Resource Center (Eugene, OR, https://zebrafish.org/home/guide.php). Zebrafish were raised in a zebTEC (Techniplast) Stand Alone Zebrafish Housing system at 28 °C[68]. The zebrafish handling was conducted according to the French and European Union guidelines for the handling of laboratory animals and approved by the local Ethics Commitee (APAFiS approval no. 2018011722529804). Young adult zebrafish 6–9 months were sampled and euthanized by immersion in a solution of tricaine methane sulfonate (MS-222, 300 mg/L) for at least 10 min. Zebrafish were dissected and organs from 20 adults (mixed male and females, except for testis and ovaries) were pooled before extraction.

**Purification of glycans**. Freshly dissected tissues were suspended in water, homogenized with a mechanical disperser and freeze dried. Dry material was sequentially extracted three times by chloroform/methanol (2:1, v/v) and chloroform/methanol (1:2, v/v). Pellets contained glycoproteins whereas supernatant contained glycolipids. Glycoprotein fraction was suspended in Tris/HCl and centrifuged at low speed. Supernatant was precipitated with 70% cold ethanol overnight and centrifuged at 1200×g. Total glycoprotein fraction was re-suspended in a solution of 6 M guanidinium chloride and 5 mM EDTA in 0.1 M Tris/HCl, pH 8, and agitated for 4 h at 4 °C. Dithiothreitol was then added to a final concentration of 20 mM and incubated for 5 h at 37 °C, followed by addition of iodoacetamide to a final concentration of 50 mM and further incubated overnight in the dark at room temperature. Reduced/alkylated sample was dialyzed against water at 4 °C for 3 days and lyophilized. The recovered protein samples were then sequentially digested by TPCK-treated trypsin for 5 h and chymotrypsin overnight at 37 °C, in 50 mM ammonium bicarbonate buffer, pH 8.4. Crude peptide fraction was separated from hydrophilic components on a C18 Sep-Pak cartridge (Waters) equilibrated in 5% acetic acid by extensive washing in the same solvent and eluted with a step gradient of 20%, 40%, and 60% propan-1-ol in 5% acetic acid. Pooled propan-1-ol fraction was dried and subjected to N-glycosidase F (Roche) digestion in 50 mM ammonium bicarbonate buffer pH 8.4, overnight at 37 °C. The released N-glycans were separated from peptides using the same C18 Sep-Pak procedure as described above. To liberate O-glycans, retained peptide fraction from C18 Sep-Pak was submitted to alkaline reductive elimination in 100 mM NaOH containing 1.0 M sodium borohydride at 37 °C for 72 h. The reaction was stopped by addition

 

of Dowex 50 × 8 cation-exchange resin (25–50 mesh, H+ form) at 4 °C until pH 6.5 and, after evaporation to dryness, boric acid was distilled as methyl ester in the presence of methanol. Total material was then submitted to cation-exchange chromatography on a Dowex 50 × 2 column (200–400 mesh, H+ form) to remove residual peptides. Glycolipid fraction was de-O-acylated with 0.1 M NaOH in chloroform/methanol (1:1, v/v) at 37 °C, 2 h and purified on a C18 Sep-Pak cartridge (Waters). Cartridge was washed with methanol/water (1:1, v/v) and eluted successively by methanol, chloroform/methanol (1:1, v/v) and (2:1, v/v). For the purification of N-glycans from brain regions, an alternative methodology based on the extraction of glycocoproteins by Triton X-100 has been used[69]. In brief, tissues were suspended in 200 μL of 1% triton in PBS and sonicated three times for 10 s. Cell debris were removed by centrifugation at 10,000×g, 4 °C for 10 min and a final concentration of 10 mM dithiothreitol and 50 mM iodoacetamide in 100 μL PBS were added sequentially to the supernatant, each followed by incubation at 37 °C for 1 h. Then 44 μL of 100% trichloroacetic acid was added and left at −20 °C for 30 min. The pellet collected by centrifugation at 10,000×g for 30 min was washed with 100% acetone three times and dried under a stream of nitrogen. Material was then subjected to trypsin and PNGAse treatment and the N-glycans purified using the conditions aforementioned.

**Structural analysis of glycans.** Following their purification, NGs, OGs, and GSLs were permethylated using the NaOH/dimethyl sulfoxide reagent[70]. The permethylated derivatives were then extracted in chloroform and repeatedly washed with water. MALDI-TOF and MALDI-TOF/TOF spectra were acquired on 4800 Proteomics Analyzer mass spectrometer (Applied Biosystems, Framingham, MA, USA) in reflecton positive or negative mode by delayed extraction using an acceleration mode of 20 kV, a pulse delay of 200 ns and grid voltage of 66%. Samples were prepared by mixing directly on the target 1 μL of oligosaccharide solution (1–5 pmol) with 1 μL of 2,5 dihydroxybenzoic acid matrix solution (10 mg/mL in $CH_3OH/H_2O$, 50/50, vol/vol). Between 50 and 100 scans were averaged for every spectrum.

The nature of each signal was established by calculating their individual monosaccharide, and optionally ceramide, contents. All identified carbohydrates-containing signals were fragmented by CID on a MALDI-TOF/TOF mass spectrometer to establish the sequences of corresponding oligosaccharides. The fragmentation patterns were all manually sorted out and eventually confirmed by automated identification tools (glycoworkbench). Only compounds presenting consistent MS/MS spectra were included into the final glycan list. In many instances, the structures of individual compounds were deduced from MS/MS spectra containing several isomers. In these cases, we confirmed the presence of a compound exclusively when a set of specific fragments could be unambiguously attributed. When needed, the structural features inferred from mass spectrometry approach were confirmed by further complementary technics including enzymatic degradation, HPLC separation of oligosialic acids, linkage analysis by GC/MS and recognition by specific antibodies. The final structural attribution took into account known biosynthetic pathways of vertebrate glycans and previous in depth studies of zebrafish glycans using MS/MS, NMR, GC/MS, and specific glycosidases[27–29]. For the sake of comparison, all glycans were relatively quantified as percentages of all identified glycan within each sample by integrating corresponding MALDI-MS signals from three independent experiments[71]. Whenever an individual signal contained several isomeric compounds, the relative proportion of each isomer was inferred from the MS/MS signal intensities of structurally related motifs.

**Analysis of sialic acids.** Intact sialic acids were released by mild hydrolysis in 0.1 N TFA at 80 °C for 1 h and coupled to DMB[29]. For DMB derivatization, sialic acids reacted with a volume of DMB reagent at 50 °C for 2 h 30 min, then separated isocratically on a C18 reverse phase HPLC column (250 × 4.6 mm, 5 μm, Vydac) by a solvent mixture of acetonitrile/methanol/water (7:9:84) and identified by referring to the elution positions of standard Neu5Ac, Neu5Gc, and Kdn derivatives[72]. Individual sialic acid derivatives were quantified by integration of fluorescence signals after HPLC separation, plotted against standard curves of corresponding authentic standards. In parallel, the nature of sialic acids was confirmed by analyzing MS and MS/MS fragmentation patterns by LC/MS based on known patterns[73,74].

**Analysis of monosaccharides.** Lipid moiety of glycolipids was analyzed by GC/MS as trimethylsilylated (TMS) derivative. Glycolipids were metanolysed (1 mL of MeOH/HCl 0.5 M) at 80 °C during 16 h. The fatty acid methyl esters (FAME) were extracted by washing the methanol fraction by heptane three times. The methanol fraction was N-re-acetylated by 20 μL of acetic anhydride then dried under nitrogen and derivatized by 30 μL BisSilylTriFluoro-Acetamide (BSTFA, Sigma-Aldrich) in 30 μL of pyridine for 2 h at room temperature before analysis by GC–MS.

**Derivatisation of lipids.** Lipid moiety of glycolipids was analyzed by GC/MS as TMS derivatives. Glycolipids were methanolysed in 500 μL of 0.5 M of 82% aq. MeOH/HCl at 80 °C for 16 h[75]. The FAME were extracted three times from the methanol fraction by 250 μL of heptane; which were pooled and concentrated to 100 μL of heptane prior to its GC/MS injections. Then, the methanol fraction was

neutralized to pH 6.5 using $Ag_2CO_3$ prior to N-re-acetylation by the addition of 20 μL of acetic anhydride at RT overnight. The methanol fraction was filtered and dried under nitrogen in order to derivatize the long chain base (LCB) by 20 μL of bissilyltrifluoroacetamide (BSTFA) in 20 μL of pyridine at RT during 2 h; prior to its GC/MS injection.

**Analysis of lipid derivatives by GC–MS.** The FAME and LCB derivatives were analyzed using a Thermo Trace Ultra gas chromatograph interfaced with a Finigann Automass II mass spectrometer. The lipid derivatives were injected (inlet temperature 280 °C) to a non-polar Alltech[R] EconoCap[TM] Capillary-Column-1 (0.25 μm film, 0.25 mm 25 m). The mass spectrometer was used in the electron ionization mode (70 eV, source temperature 150 °C) mode with automatic ionization to produce a total ion chromatogram (TIC). The FAME temperature program was 120–280 °C at 5 °C/min and 10 min left at 280 °C. The LCB-TMS temperature program was 120–240 °C at 2 °C/min, 10 min left at 240, 240– 280 °C at 4 °C/min and 10 min left at 280 °C.

**Histochemistry.** Tissues were excised from the zebrafish after fasting, washed with PBS, and fixed with 4% PFA for overnight. After washing with PBS, the tissues were serially soaked into 5%, 10%, and 15% sucrose in PBS for 1 h each and finally into 20% sucrose in PBS for overnight. They were then incubated with the OCT solution (OCT compound: 20% sucrose in PBS = 1:1) for 1 h. All procedures were performed at 4 °C. The tissues were soaked into the OCT compound and frozen at −80 °C. The frozen tissues were sliced into sections (14 μm thick), put onto the slide glass, and dried at 37 °C. The dried sections were washed five times with PBS and blocked with 5% goat serum in PBS for 1 h. Then the sections were incubated with primary antibody KDN3G[76] (1:100 dilution, 10 μg/mL) at 4 °C for overnight. After washing five times with PBS for 15 min each, they were incubated with secondary antibody (Alexa-488-conjugated anti-mouse IgG, 1:400 dilution; Thermo Fischer Scientific Ref A11001) at 4 °C for 1 h, followed by washing six times with PBS for 15 min each. Nuclear staining was performed by incubating with DAPI solution (1 μg/mL, 37 °C for 15 min). The sections were finally washed with PBS, and covered by cover glass with a drop of mounting fluid. Observation was performed under a fluoromicroscope (BX51, Olympus) or confocal scanning fluorescent microscope (LSAM 710; Karl Zeiss).

**Western blots.** Tissues were homogenized with PBS containing 1% Triton X-100, 1 μg/mL each of aprotinin, leupeptin, and pepstatin, 2 μg/mL of antipain, and 5 mM EDTA and incubated on ice for 1 h. After centrifugation of the homogenates, the protein concentration of the supernatants was measured by the bicinchoninic acid (BCA) assay. The supernatants were dissolved in Laemmli buffer containing 5% mercaptoethanol and incubated at 60 °C for 20 min. The samples were subjected to SDS–PAGE (10% or 7.5% polyacrylamide gels), followed by CBB staining and Western blotting using PVDF membranes. For immunostaining, PVDF membranes were blocked with PBS containing 0.05% Tween 20 (PBST) and 1% BSA at 25 °C for 1 h, and incubated overnight with or without the primary antibody, KDN3G (1: 1000 dilution, 10 μg/mL) at 4 °C overnight, followed by incubation with the secondary antibody, peroxidase-conjugated anti-mouse IgG + IgM (1:5000 dilution, 0.4 μg/mL; American Qualex, Ref A108PS) at 37 °C for 45 min. Visualization was performed with chemiluminescent reagents (GE Healthcare).

**RNA extraction, cDNA synthesis, and RT-PCR analysis.** Total RNA was extracted from zebrafish tissues using the Nucleospin® RNA II kit (Macherey-Nagel, Hoerdt, France). A proteinase K digestion step (55 °C, 10 min) and phenol/chloroform extraction were inserted in the protocol after dounce homogenization of the tissues and before column purification of total RNA. RNA integrity was further assessed using the RNA 6000 Nano LabChip Kit on an Agilent Bioanalyzer (Agilent Technologies, Stratagene, La Jolla, CA). For subsequent PCR amplifications, first strand cDNA was synthesized from total RNA using an oligo(dT) primer and the AffinityScript Q-PCR cDNA synthesis kit (Agilent technologies) according to the manufacturer protocol. Oligonucleotide primers were designed (Eurogentec, Herstal, Belgium) in the open-reading frame of the zebrafish sequences (Supplementary Data 5). PCR amplifications were carried out for 35 cycles with the Taq Core kit DNA polymerase (Qiagen, Courtaboeuf, France) with annealing temperatures ranging from 48 to 68 °C. Amplified fragments were subjected to 2% agarose gel electrophoresis, visualized by ethidium bromide, gel-extracted and subcloned in the pCR®2.1-TOPO vector and nucleotide sequences were confirmed by sequencing (Genoscreen, Lille, France). The uncropped agarose gel electrophoresis are shown in Supplementary Figure 8.

## Data availability

All the identified NGs, OGs and GSLs were included in Glycome Atlas that is freely available at http://www.rings.t.soka.ac.jp/GlycomeAtlasV5/index.html. The data that support the findings of this study are available from the corresponding author upon request.

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

## Acknowledgements

This work was supported by the Centre National de La Recherche Scientifique (CNRS), the ANR-2010-BLAN-120401 grant (project Galfish) from the Agence Nationale de la Recherche, SPS Strategic Young Researcher Overseas Visits Program for Accelerating Brain Circulation G2205 (to K.K.) and Mizutani Foundation for Glycoscience (to Y.G.). We are indebted to the Plateforme d'Analyse des Glycoconjugués (PAGés, http://plateforme-pages.univ-lille1.fr/) and to the Research Federation FRABio (Univ.Lille, CNRS, FR 3688, FRABio, Biochimie Structurale et Fonctionnelle des assemblages Biomoléculaires) for providing the scientific and technical environment conducive to achieving this work.

## Author contributions

N.Y., J.V., L.Y.C., L.C., S.Y.Y., J.K., and Y.G. performed the experiments; A.H.L., K.A.K., C.S., K.H.K., K.K., and Y.G. designed the study and wrote the article, all authors critically reviewed the article and approved the final manuscript.
