## [Peer Review File · Nature Communications]

Reviewers' comments:

Reviewer #1 (Remarks to the Author):

Comment and Summary: The zebrafish offers a number of advantages to study human disease mechanisms and therapeutic strategies because it is a well characterized developmental and genetic model system. Many rare genetic human diseases have been discovered in the past decade and a substantial number of these are characterized by altered glycosylation machinery. Thus, aberrant glycosylation, which has been found in cancers, inflammatory bowel diseases, neurodegenerative disorders and infectious diseases, could also contribute to an even larger fraction of currently undiagnosed human diseases. Therefore, it is very important to develop comprehensive datasets of glycan expression in the zebrafish model to provide a baseline for modeling the glycomic impact of human disease models.

Complex glycans are linked to lipids and proteins and contribute to intra- and intercellular recognition and signaling events that influence many biological functions. This manuscript describes comprehensive analysis of zebrafish glycome and sialome, including protein-linked glycan (N-glycans, NGs; and O-glycans, OGs) as well as lipid-linked glycans (glycosphingolipids, GSLs) by MALDI-TOF MS and of glycan synthetic gene expression by RT-PCR. The authors have carried out in-depth MS-based glycomic analysis to profile NGs, OGs and GSLs in adult zebrafish organs which were dissected and separated into eight classes, skin, liver, gill, brain, intestine, heart, testis and ovary. In addition, the authors describe the N-glycome in 6 regions of the zebrafish brain. The authors have also examined expression of sialyltransferases (ST), sialidases (Neu), and sialic acid synthases such as CMAS and CMAH by RT-PCR in the eight organs. Their multidimensional glycomic strategies have produced an extensive amount of glycomic data, which will be very useful for modeling human diseases associated with altered glycosylation. Their experimental approaches are comprehensive and the MS data is very solid, including MS/MS data. However, the presentation of the MS results may be difficult to comprehend for a broad audience that includes readers unfamiliar with glycomic data, possibly raising difficulties for comprehending the tissue specific expression of glycans and their sialylation patterns in zebrafish organs. Thus, the authors should strive to modify their data presentation to emphasize important differences and to make their conclusions more accessible to non-experts. Effective presentations might include heat map or clustering glycan classes to demonstrate tissue specific glycosylation patterns, rather than just showing MS profiles. In addition, the authors should describe how their data can be accessed by the broader glycoscience community. It is unclear whether the authors intend to abide by community established standards for glycan database submission or data accessibility.

Recommendation: Although the authors need to improve figures, tables and text, the quality of this research meets the journal's criteria and the extremely valuable glycomic data will be very useful for our scientific communities if the authors define how they intend to share their data with the glycoscience community. Accept with following revision.

Major Comments:

The manuscript is well-written and the data is solid. However, the data presentation could overwhelm non-glyco readers to see all of the glycomic MS data, which may not be necessary to present in this manuscript. In addition, the MS data figures are too small to see. The authors are encouraged to develop a better way for presenting and summarizing their extensive glycomic data in a manner that will be accessible for readers less familiar with glycomic data.

Figure 1 and monosaccharide symbol nomenclatures

There is no explanation about the monosaccharide symbols in the manuscript text. The color format of sialic acids and other monosaccharide species needs to be adjusted to the CFG glycan structure nomenclature. All figures for manuscript and supplementary data need to be fixed.

Figure 2

Is n=30 for NGs, OGs, and GSLs?

The authors have analyzed NGs, OGs and GSLs from eight organs, but how about glycan expression patterns in whole adult zebrafish? By comparing zebrafish embryonic glycome, is there any significant difference in the adult zebrafish glycome?

Figure 2

The authors described that the increase of Kdn intestine is not due to bacterial derived Kdn glycans as shown in Figure 9. The flow of the data would be better if figure 9 was used as figure 3 so that readers can easily see the story to appreciate that the intestinal expression of Kdn is not a bacterial byproduct.

Figure 2, 9

Although the lumen of the intestine was clearly stained by Kdn specific antibodies, I wonder if the bacterial Kdn synthase can modify intestinal glycan epitope on gut cell surface mucin to form Kdn O-glycan. The authors should discuss the possibility that bacterial sialyltransferases may be able to transfer SA onto host glycans during infection.

Figure 3 and 5 (NGs and OGs)

Have the authors analyzed sulfated NGs and OGs in zebrafish organs?

Figures 3, 5, 7

The authors crowd 8 MS spectra in one figure panel, ensuring that it is very difficult to see the annotated glycan structures. The authors need to improve their presentation of annotated MS data.

Figure 5

Is there any O-Man, O-Fuc, O-Glc type O-glycans detected in zebrafish organs?

Figure 5, m/z 1099 observed in intestine

The mass corresponds to SA1HexNAc1Hex2. Is there any MS2 data that would allow the authors to distinguish whether this is an O-GalNAc or O-Man initiated structure? There is no m/z 1099 in supplementary table 2 (OGs).

Figure 8 B

What does COH stand for?

M&M

Analysis of lipids

The authors used methanolysis for the analysis of fatty acid and sphingoid base. By using this method, fatty acid can be analyzed as methyl ester. However, for the analysis of sphingoid base, anhydrous methanolysis tends to produce 3-OMe sphingoid base. Therefore, a solvent system of Water-MeOH-HCl is commonly used for methanolysis to prepare sphingoid base to avoid the formation of 3-OMe sphingoid base. The method text does not describe what condition was used for methanolysis and a useful citation is not provided. The authors need to clarify their experimental approach. The authors also need to describe the GC column used for the analysis or add reference.

Supplemental Figure 1,2 and Table 1-3

These data are very solid but the impact of the presentation could be enhanced by including hierarchical clustering of glycan species.

Supplemental Table 3 (4/7-7/7) for GSL

There is a column showing "mass." It is assumed that "mass" indicates masses of ceramide

species but the authors need to explicitly describe this in the table as well as in the figure legend.

Supplementary Figure 9

The terminal GlcNAc can be on the 3 or 6 arm Man, in addition to possibly being a bisecting GlcNAc. Do the authors have fragmentation data supporting the assignment as bisecting GlcNAc, such as 3 substituted-Hex-HexNAc- (with Na)? The NGs #74 and #78 are only detected in intestine. Can the authors include an additional experiment to confirm the presence of bisecting GlcNAc in intestine by staining with EPHA or other lectins that specifically recognize the bisecting GlcNAc.

Supplementary figure 5, 7-10, 13, 14

Have the authors analyzed glycan samples under conditions that drive the predominant production of sodiated adducts in order to simplify the MSn pattern? The presented data is complicated by the mixture of glycans with Na and H derived fragment ions.

Supplementary Figure 3

This data should be presented more effectively and included in the manuscript, not placed in the supplement.

Minor Comments:

The authors need to check their manuscript text carefully because there are still considerable amounts of typos written in French.

There are a lot of typos in monosaccharide description such as GlcNAc, Glnac, GlcNac, therefore, the authors need to check and fix them.

The texts in manuscript and supplementary notes need to be polished because there is too many information that may not be necessary to describe the detail.

Reviewer #2 (Remarks to the Author):

The manuscript provides the first complete glycome mapping of adult zebrafish, an essential animal model to study the glycome in the context of health and disease. They used a combination of analytical techniques to unravel the glycome and the glycosyltransferases present in adult zebrafish. This work is of high relevance to any research scientist interested in glycosylation.

Major point to revise:

Glycans were permethylated using methyl iodide, which doesn't allow studying the kind of sialic acid linkage. Why do authors only assign sialic acids to be only 2-3-linked? Hanzawa et al., (Glycobiology. 2017 Mar 4;27(3):228-245.) recently showed the presence of fair amounts of 2-6 linked sialic acids (Neu5Ac) in zebrafish embryos. How come they would have disappeared in adults?

Minor points:

Figure 2: what part of the graph was made using NP-HPLC-FL measurements? Same question using LC-MS

Supplementary Figures 1 and 2: glycan structures are too small to be seen even when zooming the document

Minor editing of the manuscript and supplementary data is needed: capital letters are at times missing: Neu5ac instead of Neu5Ac, etc...

Reviewer #3 (Remarks to the Author):

Owing to its advantages (e.g., gene tractability, transparency, etc.), zebrafish (*Danio rerio*) has emerged as a popular model organism for high-throughput study of many biological processes and human diseases. These features also have made zebrafish a good mode to study glycosylation. Dr. Guérardel and colleagues collected eight organs from zebrafishes and performed the standard isolation of N-linked glycans, O-linked glycans, and glycosphingolipids. Differences in sialic acids composition in different organs were revealed by DMB derivation assisted HPLC quantification. The authors also conducted MS assays (e.g., MALDI-TOF, MS/MS, GC/MS) to profile glycoforms in different tissues. These data provided abundant information related to glycosylation (e.g., fucosylation, sialylation) in different organs, including relative abundance of a particular type of glycans and organ-specific glycans. Those observations should be a very important guide for studies of zebrafish glycosylation. This method can be easily expanded to profile more sophisticated structures such as different brain regions as shown in this manuscript.

While I am very much excited about this manuscript for the above reasons, the following concerns must be addressed:

1. For DMB derivation based quantification of sialic acids in different tissues, how many biological samples (repeats) were evaluated here (Figure 2)? Quantity of sialic acids was presented as nmol/mg protein. How accurate are these data? It is like comparing the weight of a few sesame seeds with the weight of a few pounds of apples. Huge errors are involved. Therefore, additional data (e.g., lectin staining of tissue slices) are required to support the observations here.
2. It is expected that different organs express different glycoforms as revealed in the MALDI-TOF results in Figure 3, 5, and 6. However, for the quantitative assays performed in this manuscript, at least three biological repeats should be conducted. In Figure 4, 6 and 8, the essential biological replicates were missed in the assay. Data presented in Figures 4, 6, 8 were aimed to count the numbers of particular types of glycoforms in the total number of glycans identified. The authors should clearly point this out in their manuscript, otherwise, it is misleading for readers that these quantifications present the actual abundance of a particular glycoform in that organ.
3. In Figure 9, the figure legend did not describe the figure clearly. For example, there was no definition of Gp-Kdn. Furthermore, there was no negative control used for Figure 9C. Missed citation for using monoclonal antibody KDN3G also should be added.
4. In Figure 10, authors did not present enough data to support their claim of structural analysis of brain regions. How many samples were analyzed? How error bars were calculated?

Overall, the author needs to be more rigorous in their analysis and include additional controls for their experiments.

Response to reviewer's comments:**Reviewer #1 (Remarks to the Author):**

1-However, the presentation of the MS results may be difficult to comprehend for a broad audience that includes readers unfamiliar with glycomic data, possibly raising difficulties for comprehending the tissue specific expression of glycans and their sialylation patterns in zebrafish organs. Thus, the authors should strive to modify their data presentation to emphasize important differences and to make their conclusions more accessible to non-experts. Effective presentations might include heat map or clustering glycan classes to demonstrate tissue specific glycosylation patterns, rather than just showing MS profiles.

We agree with the reviewer that the data are not easy to access considering the large number of glycans that have been described. In order to emphasize the organ-specific differences and focus on the glycans motifs rather than on the individual structures, we have largely followed the Reviewer's advice and presented the data in the main manuscript as heat maps (Fig 3, 5, 6). The glycans have been clustered according to the nature of terminal motifs, sialic acids and epitopes. The values in the heat maps were extracted from relative intensities of O-glycans and N-glycans MS signals (three replicates). When isomers were present, values were calculated according to the intensities MS/MS signals. For glycolipids, we did not include values because the lipid heterogeneity prevents precise signal integration. It was decided not to use endo-ceramidase due to the known specificities of this enzyme. In addition, all the MS spectra shown in the initial version of the manuscript (former Fig 3, 5, 7) are shown in a larger format in the supplementary data.

2- In addition, the authors should describe how their data can be accessed by the broader glycoscience community. It is unclear whether the authors intend to abide by community established standards for glycan database submission or data accessibility.

Initially, we intended to develop a database dedicated to zebrafish glycosylation to provide an easy access to our data. However, we finally decided to implement our data in existing glycoscience databases that are already accessible to the glycoscience community. As a first step, we are developing in collaboration with Prof Kiyoko Aoki-Kinoshita a new interface to the Glycome Atlas initiative in order to expand it toward zebrafish. Presently, this web tool allows users to visualize and perform organ and species-specific queries of glycome data in Human and mouse (<http://www.rings.t.soka.ac.jp/GlycomeAtlas/GUI.html>) in an interactive manner. To accomplish this, we have coded all identified zebrafish glycans using published linear coding, as described by Banin and collaborators, *TiGG* 2002, 14, pp127-137. Then the team of Prof Aoki-Kinoshita have designed a zebrafish-specific interface and implement the coded glycans according to Supplementary Tables 1 (NGs), 2 (OGs) and 3 (glycan moieties of GSLs). This data base is presently accessible at <http://rings.t.soka.ac.jp/GlycomeAtlasV5/index.html>. As a second step, we will implement our data in UnicarbKB database that permits global glycan search.

3- The manuscript is well-written and the data is solid. However, the data presentation could overwhelm non-glyco readers to see all of the glycomic MS data, which may not be necessary to present in this manuscript. In addition, the MS data figures are too small to see. The authors are encouraged to develop a better way for presenting and summarizing their extensive glycomic data in a manner that will be accessible for readers less familiar with glycomic data.

See point 1

4- Figure 1 and monosaccharide symbol nomenclatures - There is no explanation about the monosaccharide symbols in the manuscript text.

The Symbols used are those recommended by SNFG. In addition to citing the references, the symbol keys are now added to the legends of Figure 1 and Figure S1. Furthermore, an abbreviation section was added in the main manuscript.

5- The color format of sialic acids and other monosaccharide species needs to be adjusted to the CFG glycan structure nomenclature. All figures for manuscript and supplementary data need to be fixed.

The reviewer is right; the colour format did not fit the recently agreed consensus for monosaccharides nomenclatures. As required by the reviewer, we have modified the colours of monosaccharides in all figures and supplementary tables based on the recent SNFG note (<https://www.ncbi.nlm.nih.gov/glycans/snfg.html#tab2>) that redefined the colour assignments. We have used the following CMYK values:

	Gal	GalNAc	Glc	GlcNAc	Man	Fuc	Neu5Ac	Neu5Gc	Kdn
									C	0	0	100	100	100	0	38	41	100
M	15	15	50	50	0	100	88	5	0
Y	100	100	0	0	100	100	0	3	100
K	0	0	0	0	0	0	0	0	0

6- Figure 2. Is n=30 for NGs, OGs, and GSLs?

No, n=3. Three separate batches of 8 individual organs pooled from 30 zebrafishes each were used as starting materials. NGs, OGs and GSLs were then extracted from the three pools and analysed in parallel for structure and sialic acid content. We decided to use pools of organs from 30 zebrafishes to obtain more robust analysis considering that individual organs are very small, which would lead to unacceptable inter-individuals variability. This is now clarified in the first sentence of the Results section.

7- The authors have analyzed NGs, OGs and GSLs from eight organs, but how about glycan expression patterns in whole adult zebrafish?

We did not perform the glycan pattern analysis on total whole adult zebrafish precisely because the profiles would have resulted from a mixture of too many different tissues and organs. This analytical approach was, however, conducted on whole zebrafish embryos at different developmental stages, as reported previously (Takemoto, 2005; Guerardel et al., 2006; Chang et al. 2009; Hanzawa et al., 2017). Here we wished to provide organ-specific patterns of glycosylation in order to dissect the mechanisms by which specific glycans segregate along organogenesis. The important structural differences noticed between organs validated ex post this approach.

8- By comparing zebrafish embryonic glycome, is there any significant difference in the adult zebrafish glycome?

Yes, we have observed numerous differences which have been highlighted in the revised version of the manuscript.

9- Figure 2 - The authors described that the increase of Kdn intestine is not due to bacterial derived Kdn glycans as shown in Figure 9. The flow of the data would be better if figure 9 was used as figure 3 so that readers can easily see the story to appreciate that the intestinal expression of Kdn is not a bacterial byproduct.

In order to answer to reviewer's query we have conducted further analyses on the possible origin of the Kdn observed in intestinal lumen and in zebrafish faeces. In particular, we have further analysed zebrafish faeces and demonstrated that Kdn was predominantly associated with the microbiota glycoconjugates, presumably polysaccharides. These data were included in a new version of the Figure 9 (Fig 8 in the revised version). However, considering the large amount of data associated with this figure, we propose to keep it in its original position.

10 -Figure 2, 9 - Although the lumen of the intestine was clearly stained by Kdn specific antibodies, I wonder if the bacterial Kdn synthase can modify intestinal glycan epitope on gut cell surface mucin to form Kdn O-glycan. The authors should discuss the possibility that bacterial sialyltransferases may be able to transfer SA onto host glycans during infection.

The reviewer's hypothesis is also very exciting and should be taken into account. At present time, we have no experimental data that could directly discriminate a glycosyltransferase bacterial activity from host. However, considering that (1) Kdn is systematically transferred in place of a Neu5Ac or a Neu5Gc that are known to be transferred by zebrafish sialyltransferases and (2) so far no study ever reported a clear specificity of a vertebrate sialyltransferase toward CMP-Neu5Ac/Neu5Gc/Kdn, we believe that the transfer results from the activity of an endogenous enzyme that can use any of the three accessible activated sialic acids. This hypothesis was mentioned in the revised version.

11 - Figure 3 and 5 (NGs and OGs) - Have the authors analyzed sulfated NGs and OGs in zebrafish organs?

Actually, we have obtained preliminary results showing the presence of sulphated mucin-type O-glycans. However, considering the large amount of data that we had to deal with and the fact that it would require entirely different extraction and analytical schemes, we have decided not to mention it in the present manuscript and not to develop a specific analytical scheme toward the analysis of sulphated glycans. We expect to develop this particular aspect in a coming report. This future report will also focus on the organ-specific expression of sulfotransferases.

12 - Figures 3, 5, 7 - The authors crowd 8 MS spectra into one figure panel, ensuring that it is very difficult to see the annotated glycan structures. The authors need to improve their presentation of annotated MS data.

We agree with the reviewer that these figures are overcrowded. Furthermore, the PDF conversion further degraded the overall quality of data. Thus, as previously mentioned, we have removed from the main manuscript most of the MS data and transferred larger versions of the spectra in supplementary material.

13 - Figure 5 - Is there any O-Man, O-Fuc, O-Glc type O-glycans detected in zebrafish organs?

We have so far not detected this type of O-linked glycosylation. All the MS/MS spectra we have analysed exclusively showed the presence of O-GalNAc type O-glycans. However, it remains probable that such forms of O-glycosylation may be present. We have added this information in the main manuscript.

14- Figure 5, m/z 1099 observed in intestine- The mass corresponds to SA1HexNAc1Hex2. Is there any MS2 data that would allow the authors to distinguish whether this is an O-GalNAc or O-Man initiated structure? There is no m/z 1099 in supplementary table 2 (OGs).

The reviewer is absolutely right about the possibility that signal at m/z 1099 may be the equivalent of O-mannosylated glycan that was previously identified in brain derived glycoproteins (Stalnakar et al., 2011). As suggested by the reviewer we have conducted MS/MS experiments in order to identify this potential O-linked glycans that can be observed as a minor component in several tissues. These data did not permit assigning the 1099 signal to any glycan. As shown here, the MS/MS pattern of m/z 1099 signal observed in intestine did not reveal any fragment typically assigned to glycan. Thus, we have concluded that this signal corresponded to non-carbohydrate yet unidentified compound and did not assigned it on the spectra.

In comparison, the MS/MS spectrum of 1099 signal assigned to **SA1HexNAc1Hex2** in the report from Stalnakar et al clearly indicates the presence of this tetrasaccharide, as mentioned by the reviewer.

MS/MS analysis of permethylated signal at m/z 1100 from LARGE^{+/+} mouse brain proteins, as seen in Stalnakar et al, 2011 J Biol Chem 286, pp21180-21190

15- Figure 8 B- What does COH stand for?

COH stands for “carbohydrate moieties”. Along the revision of the manuscript, we have deleted the panel 8B that was somehow redundant with supplementary tables.

16 - Analysis of lipids. The authors used methanolysis for the analysis of fatty acid and sphingoid base. By using this method, fatty acid can be analyzed as methyl ester. However, for the analysis of sphingoid base, anhydrous methanolysis tends to produce 3-OMe sphingoid base. Therefore, a solvent system of Water-MeOH-HCl is commonly used for methanolysis to prepare sphingoid base to avoid the formation of 3-OMe sphingoid base. The method text does not describe what condition was used for methanolysis and a useful citation is not provided. The authors need to clarify their experimental approach. The authors also need to describe the GC column used for the analysis or add reference.

We thank the reviewer for the comment. Indeed, we did not include a dedicated experimental section for the analysis of sphingosines. As instructed, we have provided further experimental details in the material and methods sections with reference, as described below. Using this methodology, our analyses did not reveal the presence of 3-OMe sphingoid base as suggested by the reviewer.

Derivatisation of lipids - Lipid moiety of glycolipids was analysed by GC/MS as trimethylsilylated (TMS) derivatives. Glycolipids were methanolysed in 500 μ L of 0.5 M of 82% aq. MeOH/HCl at 80°C for 16 h (Gaver and Sweeley, 1965). The fatty acid methyl esters (FAME) were extracted three times from the methanol fraction by 250 μ L of heptane; which were pooled and concentrated to 100 μ L of heptane prior to its GC/MS injections. Then, the methanol fraction was neutralised to pH 6.5 using Ag_2CO_3 prior to *N*-re-acetylation by the addition of 20 μ L of acetic anhydride at RT overnight. The methanol fraction was filtered and dried under nitrogen in order to derivatize the long chain base (LCB) by 20 μ L of bis(trimethylsilyl)trifluoroacetamide (BSTFA) in 20 μ L of pyridine at RT during 2 h; prior to its GC/MS injection.

Analysis of lipid derivatives by GC-MS - The FAME and LCB derivatives were analysed using a Thermo Trace Ultra gas chromatograph interfaced with a Finigann Automass II mass spectrometer. The lipid

derivatives were injected (inlet temperature 280°C) to a non-polar Alltech^R EconoCapTM Capillary-Column-1 (0.25 µm film, 0.25 mm 25 m). The mass spectrometer was used in the electron ionization mode (70 eV, source temperature 150°C) mode with automatic ionization to produce a total ion chromatogram (TIC). The FAME temperature program was 120°C to 280°C at 5°C/min and 10 min left at 280°C. The LCB-TMS temperature program was 120°C to 240°C at 2°C/min, 10 min left at 240°C, 240°C to 280°C at 4°C/min and 10min left at 280°C.

Reference

Gaver RC, Sweeley CC. 1965. Methods for Methanolysis of Sphingolipids and Direct Determination of Long-Chain Bases by Gas Chromatography. J Am Oil Chem Soc 42:294-8.

17- Supplemental Figure 1,2 and Table 1-3- These data are very solid but the impact of the presentation could be enhanced by including hierarchical clustering of glycan species.

As afore mentioned, we have included clustering of glycan species in heat maps for NGs, OGs and GSLs into the main manuscript. Thus Sup Fig. 1,2 and Tables will come as support for these data.

18- Supplemental Table 3 (4/7-7/7) for GSL. There is a column showing “mass.” It is assumed that “mass” indicates masses of ceramide species but the authors need to explicitly describe this in the table as well as in the figure legend.

Values in “mass” column correspond to the diagnostic Z fragmentation ions for the methylated ceramide moiety. Most probable combinations of sphingoid bases and fatty acids based on composition and Z fragment ions are provided in the “SB, FA” column. This information have been added in the Supplementary Table 3.

19- Supplementary Figure 9- the terminal GlcNAc can be on the 3 or 6 arm Man, in addition to possibly being a bisecting GlcNAc. Do the authors have fragmentation data supporting the assignment as bisecting GlcNAc, such as 3 substituted-Hex-HexNAc- (with Na)?

All MS/MS fragmentation spectra of NGs carrying an extra HexNAc support the presence of a single GlcNAc that substitutes the internal Man residue (bisecting GlcNAc) or alternatively the 3-/6-arm Man. Indeed, no B/Y ion supports the presence of HexNAc extended LacNAc. This is exemplified in Suppl Fig 14 and further developed in the spectra below. Some structures possess a single extra GlcNAc (A & B), presumably a bisecting GlcNAc, whereas others present two (C & D), presumably one as bisecting and one on the branch (either 3- or 6- arm).

A**B****C**
D

However, differentiating the position of single GlcNAc residues either on internal Man or 3-/6-arm Man was not possible from our MS/MS experiments. Considering that tri- and tetra-antennary NGs have only been detected as traces in all organs, we considered the extra GlcNAc residues as most likely to be bisecting. This line of reasoning has now been included.

20- The NGs #74 and #78 are only detected in intestine. Can the authors include an additional experiment to confirm the presence of bisecting GlcNAc in intestine by staining with EPHA or other lectins that specifically recognize the bisecting GlcNAc.

The reviewer is right about the specificity of NG 74 and 78. However, it seems difficult even impossible to confirm the specificity of bisecting GlcNAc by lectin staining because bisecting GlcNAc is ubiquitously distributed in all tissues. So we are afraid that lectins may not be able to discriminate any particular structure, including 74 and 78. Nonetheless, it seems clear that bisecting GlcNAc is not homogeneously spread in all organs. Indeed, in most organs (5 out of 8), bisecting GlcNAc substitute from 36 to 45% of NGs. This ratio is lower in skin (25%) and ovary (16%) and very low in brain (10%). This information that was not initially mentioned into the manuscript was added.

21- Supplementary figure 5, 7-10, 13, 14. Have the authors analyzed glycan samples under conditions that drive the predominant production of sodiated adducts in order to simplify the MSn pattern? The presented data is complicated by the mixture of glycans with Na and H derived fragment ions.

No, we have not used specific conditions. All MALDI-MS analyses were acquired using standard procedures that naturally result in predominant production of sodiated adducts in positive mode for both MS and MS/MS experiment. As a rule of thumb, MALDI-MS spectra for permethylated glycans of sodiated parent only gives sodiated fragment ions via distinctive cleavages. In our case, the only exception is the non-sodiated B ions corresponding to the oxonium ions of single sialic acids. However, their low m/z value never interfered with our analyses and thus we did not attempt to get rid of it. This has been described in the Supplementary Data section of the revised manuscript.

22- Supplementary Figure 3- This data should be presented more effectively and included in the manuscript, not placed in the supplement.

As proposed by the reviewer, we have put more emphasis on the glycosyltransferases transcript data in the main text by inserting the previous Sup Figure 3 in the main manuscript as a heatmap (Figure 7) and keeping the original data as Supplementary Figure 7.

Minor Comments:

23- The authors need to check their manuscript text carefully because there are still considerable amounts of typos written in French.

The revised version has been carefully proof read and corrected by an English native speaker.

24- There are a lot of typos in monosaccharide description such as GlcNAc, Glnac, GlcNac, therefore, the authors need to check and fix them.

These mistakes have been corrected throughout the manuscript.

25- The texts in manuscript and supplementary notes need to be polished because there is too many information that may not be necessary to describe the detail.

As advised by the reviewer, we have considerably reduced the number of technical information in the main manuscript, but add more elements about discussion.

Reviewer #2 (Remarks to the Author):

Major point to revise:

26- Glycans were permethylated using methyl iodide, which doesn't allow studying the kind of sialic acid linkage. Why do authors only assign sialic acids to be only 2-3-linked? Hanzawa et al., (Glycobiology. 2017 Mar 4;27(3):228-245.) recently showed the presence of fair amounts of 2-6 linked sialic acids (Neu5Ac) in zebrafish embryos. How come they would have disappeared in adults?

We agree with the reviewer that the sialic acid α 2-6/2-3 linkage is an important issue that should have been discussed further in the manuscript. As mentioned by the reviewer, Hanzawa et al. convincingly demonstrated that N-glycans α 2,6-substituted on Gal were the predominantly expressed sialylated N-glycans in devolged embryos. In particular, mono- and di-sialylated bi-antennary N-glycans represented the major embryonic N-glycans. These glycans appeared to be either absent or very minor N-glycans in all adult organs. Indeed, only mono-sialylated di-antennary NG20 could be observed at the MS level in gills and represented less than 5% of complex N-glycans, whereas its di-sialylated equivalent could not be observed in any tissue. Based on Hanzawa observations, the only potentially α 2,6-di-sialylated NG on galactose that could be observed at MS level in adult is core-fucosylated di-sialylated NG60 that represents less than 10% of complex NGs in brain. In addition to the Galactose-sialylation, we have identified in the present study a number of LacdiNac N-glycans that are substituted exclusively on the terminal GalNAc residue. According to Hanzawa and

collaborators, sialic acid substitute the GalNAc residue exclusively in a2-6 position. These glycans were surprisingly exclusively observed in ovary. Thus altogether, we can conclude that the major sialylated N-glycans observed in embryos are very minor glycans in adult. Contrarily, most of the sialylated N-glycans that we have identified in adults are derived from those carrying a2,3-substituted zebrafish epitope $\text{Gal}(\beta 1,4)_{0-1}(\text{NeuAc}/\text{Gc}(\alpha 2,3))\text{Gal}(\beta 1,4)[\text{Fuc}(\alpha 1,3)]_{0-1}\text{GlcNAc}$ motif which strongly suggests that sialylated epitopes are shifting from a2-6 to a2-3 from embryos to adulthood. This hypothesis is in agreement with two observations from Hanzawa and collaborators who showed that (1) adult derived yolk does not contain a2-6 linked sialic acid and (2) that the a2-6 sialylation strongly decreases along embryonic development from 6 to 48 hpf whereas the a2-3 sialylation increases. However, despite the compelling evidences of a shift toward a2-3 sialylation, our MS/MS data indeed could not discriminate a2,3 from a2,6. We thus cannot preclude the possibility that minor sialylated NGs may be a2,6 sialylated galactose without totally redesigning the experimental procedure. These aspects have now been further elaborated in the revised version of the manuscript.

27- Minor points: Figure 2: what part of the graph was made using NP-HPLC-FL measurements? Same question using LC-MS

We have used fluorescence detection after HPLC separation to quantify the sialic acid derivatives and LC-MS/MS to confirm the attribution and quantifications. The data shown in Figure 2 is using HPLC-FL for all quantification. This has now been clarified in the revised version of the Methods sections.

28 -Supplementary Figures 1 and 2: glycan structures are too small to be seen oven when zooming the document

The glycan structures have been rearranged and their size increased.

29- Minor editing of the manuscript and supplementary data is needed: capital letters are at times missing: Neu5ac instead of Neu5Ac, etc...

As suggested by the reviewer, we have proof read the manuscript and corrected the typos.

Reviewer #3 (Remarks to the Author):

30. For DMB derivation based quantification of sialic acids in different tissues, how many biological samples (repeats) were evaluated here (Figure 2)?

Quantification values and standard deviations were based on three biological repeats.

31. Quantity of sialic acids was presented as nmol/mg protein. How accurate are these data? It is like comparing the weight of a few sesame seeds with the weight of a few pounds of apples. Huge errors are involved.

We agree with the reviewer that the reported quantities of sialic acids, ranging from about 0.6 to 41 nmol/mg, with a gross average around 5 nmol/mg, may seem rather low. However, when considered in weight, these values appear to be fairly high, ranging from about 0.18 up to 12 ug per mg of dry tissues. These values are in total agreement with those previously reported in different models, using different methodologies, by a number of researchers including Roland Schauer or Kazuaki Kakehi.

More precisely, previous reports showed that rainbow trout gametes contains 1.95 nmol/mg of proteins (Angata et al. 1999), loach testis 4.03 nmol/mg (Yasukawa et al 2009) mice tissues 0.4 to 1.1 nmol/mg (go et al. 2006) and B16 mouse melanoma cells 14.2 nmol/mg (Angata et al., BBRC 1999). This data have been added to the manuscript.

DMB-derivatization coupled to HPLC-FL appears to be the most robust, sensitive and specific approach to quantify sialic acid from limited amounts of total tissues and purified glycans, especially when it is confirmed by LC-MS. Nowadays, few other method can address this difficult problem. They include direct quantification by ion exchange chromatography and gas chromatography following sialic acid release, but both methods lack the sensitivity of fluorescence detection and are based on the same general principles. However, we agree with the reviewer that one should challenge the robustness of DMB derivation based quantification by a totally orthogonal methodology.

In order to go down this road, we have initiated the development of a novel methodology to quantify sialic acids that is totally independent from their chemical or enzymatic release. It uses nanoLC-MS²/MS³ Glycotope centric analysis of permethylated glycans on Orbitrap Fusion Tribrid. Indeed, we have recently demonstrated that this analytical workflow enabled a reliable relative quantification of specific glycotopes independently of the nature of the carrying N- or O-glycans (Hsiao CT et al, Mol cell Proteomics PMID: 29066631). The relative abundance of a given glycotope as defined by a diagnostic MS² ion can be correlated with the number of MS² spectra in which its identifying diagnostic MS² ion is detected within 5 ppm mass accuracy, and/or the sum of its peak intensity. We used this as an orthogonal approach to validate the DMB-based quantification of sialic acid species associated with N-glycans by selecting MS² oxonium ions to identify sialylated epitopes as Sia₁HexNAC₂⁺ (*m/z* 866/896 for when Sia=Neu5Ac/Neu5Gc), Sia₁Hex₁HexNAC₁⁺ (*m/z* 825/855/784 when Sia=Neu5Ac/Neu5Gc/Kdn), Sia₁Hex₂HexNAC₂⁺ (*m/z* 1274 when Sia=Neu5Ac), Sia₁deHex₁Hex₂HexNAC₂⁺ (*m/z* 1448/1478 when Sia=Neu5Ac/Neu5Gc). pd-MS³ acquisition was then used to confirm the nature of sialic acid (*m/z* 374/344/271 for neu5Ac/Neu5Gc/Kdn) associated with the MS² detected target epitopes. Then, the ion intensities of all MS² ions for each sialylated epitopes, depending on the nature of substituting sialic acid, were summed using data mining computational tool GlyPick and plotted in order to compare their relative intensities.

Preliminary results based on the analysis of three zebrafish tissues showed relative ratios of sialic acids very similar to those established by NP-HPLC-FL. Although this methodology cannot provide a true quantification of sialic acid as NP-HPLC-FL does, these preliminary experiments strongly support the results obtained by NP-HPLC-FL, in agreement with the abundant literature that used NP-HPLC-FL as a valid method of quantification. As this methodology is not yet fully developed and will be more thoroughly explored and applied in a coming report, we opted not to mention these in the final manuscript.

32. Therefore, additional data (e.g., lectin staining of tissue slices) are required to support the observations here.

We definitely agree that lectin staining on tissue section would be very interesting and bring further insight into the fine localisation of glycan epitopes and should be explored in zebrafish. However, in the specific context of sialic acids quantification, it appears that it would bring no further information as we do not have access to specific lectins that can recognize and differentially quantify Neu5Ac, Neu5Gc and Kdn independently of the underlying glycan epitopes. Even if we had access to these

reagents, one unfortunately cannot quantify NeuAc/NeuGc/Kdn based on differential lectin/mAb staining. We are presently trying to develop an alternative methodology for spatial resolution of different sialic acids using MALDI imaging technology as an alternative to lectin and antibody tissue staining and hope we will be soon able to share these results.

33. It is expected that different organs express different glycoforms as revealed in the MALDI-TOF results in Figure 3, 5, and 6. However, for the quantitative assays performed in this manuscript, at least three biological repeats should be conducted.

The quantification data provides in Sup Fig 2 and 3 for N-glycans and O-glycans result from three independent experiments, as suggested by the reviewer. This information is already included in the Methods section, but we have made this point clearer in the revised version of the manuscript.

34. In Figure 4, 6 and 8, the essential biological replicates were missed in the assay.

The Reviewer may have been misled into assuming that no biological replicate was included in Fig 4, 6 and 8. As already mentioned in answer to point 6 above, the 8 individual organs derived from 3 separate batches of 30 zebrafishes each were pooled as starting sample materials for all glycomic analysis. In fact, the numbers and % in Fig. 4, 6 and 8 were established by counting the confirmed glycans substituted by one or another epitope (Supplementary Tables 1, 2, 3 and 4). To be included into Sup Table 1-4, MS signal of individual glycans had to be observed in all three biological replicates and confirmed by MS/MS analysis. We deliberately chose to ignore inconsistently detected or uncertified signals. As a result, the compiled individual data in former Fig 4, 6 and 8 (now 4, 5 and 6) already took into account the results from three independent experiments and the glycan numbers referred to the total glycan species commonly detected in a particular organ derived from all 3 biological replicates, which itself comprised a pool of that specific organ dissected from 30 different zebrafishes each.

35. Data presented in Figures 4, 6, 8 were aimed to count the numbers of particular types of glycoforms in the total number of glycans identified. The authors should clearly point this out in their manuscript, Otherwise, It is a misleading for readers that these quantifications presents the actual abundance of a particular glycoform in that organ.

We agree with the reviewer that mixing data from counting the distinct glycan numbers and those from relative quantifications based on integration of their peak intensities, would be misleading. We have modified the presentation of the figures and make things clearer in the revised version.

36. In Figure 9, the figure legend did not describe the figure clearly. For example, there was no definition of Gp-Kdn. Furthermore, there was no negative control used for Figure 9C. Missed citation for using monoclonal antibody KDN3G also should be added.

We agree with the reviewer that some information is missing. To answer to reviewer's comment, we have (1) redesigned the former Fig. 9 (now Fig. 8) to focus on meaningful data; (2) added new data to clarify the origin of Kdn (Fig. 8C); (3) added an illustration to explain our hypothesis (Fig. 8D); (4) modified the associated text in the main manuscript; (5) transferred control experiments for WB in supplementary data (Supplementary Fig. 8); (6) added immunochemistry controls in Supplementary data (Supplementary Fig. 9); (7) added information and references about Gp-Kdn and KDN3G in the legend of Figure 8.

37. In Figure 10, authors did not present enough data to support their claim of structural analysis of brain regions. How many samples were analyzed? How error bars were calculated?

We agree with the reviewer that the analysis of brain region was not supported by enough data. In particular, the N-glycans profiling was established from a single batch of brain regions, which does not provide meaningful enough information. In order to answer to this query, we have conducted two sets of experiments. On one hand, we have optimized the release and the analysis of NGs from each region in order to improve the quality of the MS spectra. In particular, considering the low quantity of starting material, we have relied on an alternative methodology based on triton X-100 extraction of glycoproteins before N-glycans release that resulted in a dramatic improvement of the spectra quality. Analysis of three independent replicates generated almost identical MS spectra. The methodology was added in the Methods section of the main manuscript and spectra of one of the three batches are provided in the Supplementary Fig. 12. On the other hand, we have replicated the sialic acid composition analyses on total brain regions using nine independent samples collected at different times in order to increase the statistical significance of the assay. The overall analysis of these data did not result in significant differences between regions, which demonstrate that our previous analysis was not robust enough. This information was also added in the manuscript.

REVIEWERS' COMMENTS:

Reviewer #1 (Remarks to the Author):

Summary: The authors have improved figures, tables, text and also developed bioinformatic tools for the zebrafish glycan expression. The manuscript has been significantly improved and revised as requested. The revised manuscript is publishable after addressing the following minor revisions.

Recommendation: Accept with following minor revisions.

Minor comment:

Reviewer 1 comment #2: The authors have developed a useful bioinformatics tool for zebrafish glycomics. The GlycomeAtlas developed by Dr. Kiyoko Kinoshita-Aoki's group originally included the structures of glycans for mice, human, and Drosophila. It is very nice that Dr. Guerardel and Dr. Kinoshita-Aoki have worked together to develop the new tool and included zebrafish glycan structures in the tools. However, this manuscript "NCOMMS-17-25015A" does not include instructions for using GlycomeAtlas or other detailed information that would allow users to take advantage of the new functions of GlycomeAtlas for visualizing zebrafish glycomes. I wonder if the authors could add text or refer to other sources that describe the improved GlycomeAtlas for zebrafish. This will be very important to do in order to provide the information to glycobiology and glycobioinformatics communities. In addition, the new GlycomeAtlas has a great function for visualizing tissue-specific glycan expression which will prove to be a very useful tool for using zebrafish as a model system for human diseases caused by altered glycosylation and will contribute significantly to future studies in biomedical research.

Reviewer 1 comment #11: Sulfated glycan and sulfotransferase. It is great to know that the authors will carry out the analysis of organ-specific expression of sulfotransferases as well as the analysis of sulfated glycans. I look forward to seeing the new manuscript in the future.

Page 34, Analysis of sialic acids. I looked at reference # 76 (Hara et al), but they used acetic acid for hydrolysis not TFA. Although this reference is suitable for citation, the authors need to include a correct reference using TFA for the hydrolysis. The method will also need to include the incubation time used for mild acid hydrolysis.

Reviewer #3 (Remarks to the Author):

In the revised manuscript, the authors have conducted additional experiments to address all major concerns. The manuscript is in good shape to be published.

REVIEWERS' COMMENTS:

Reviewer #1 (Remarks to the Author):

Minor comment:

The authors have developed a useful bioinformatics tool for zebrafish glycomics. The GlycomeAtlas developed by Dr. Kiyoko Kinoshita-Aoki's group originally included the structures of glycans for mice, human, and Drosophila. It is very nice that Dr. Guerardel and Dr. Kinoshita-Aoki have worked together to develop the new tool and included zebrafish glycan structures in the tools.

However, this manuscript "NCOMMS-17-25015A" does not include instructions for using GlycomeAtlas or other detailed information that would allow users to take advantage of the new functions of GlycomeAtlas for visualizing zebrafish glycomes. I wonder if the authors could add text or refer to other sources that describe the improved GlycomeAtlas for zebrafish. This will be very important to do in order to provide the information to glycobiology and glycoinformatics communities. In addition, the new GlycomeAtlas has a great function for visualizing tissue-specific glycan expression which will prove to be a very useful tool for using zebrafish as a model system for human diseases caused by altered glycosylation and will contribute significantly to future studies in biomedical research.

- *As proposed by the reviewer, we have now included in the results section of the manuscript a chapter that describes instruction for the use of Glycome Atlas as follows:*

To use the updated GlycomeAtlas version 5, three buttons are available at the top for "human", "mouse" and now "zebrafish". By clicking on "zebrafish", an image of the zebrafish with organs outlined and a table of the eight organs for which glycans have been characterized will be displayed. By clicking on either the image or the table, the selected organ and row in the table will be highlighted in yellow, and the list of glycans profiled will be shown on the right. When a single glycan is clicked in this area, it will be highlighted and its detailed structure will be displayed below the image. Clicking again on another glycan will add the glycan to the selection list below the image. At the same time, the other organs in which the selected glycan(s) are found will also be highlighted in both the image and the table. This allows users to see the tissue-specificity of the glycan(s) selected. Right-clicking on a selected glycan will display a pop-up menu for three options: 1) to copy the LinearCode® format for the glycan, which can be used for Glycan Search (available in the menu on the left), 2) open a window to the CFG if the glycan data is available there, or 3) open a window to GlyTouCan, if the glycan data is available.⁴² The selected glycans can be deselected by clicking the "x" on the upper right.

Page 34, Analysis of sialic acids. I looked at reference # 76 (Hara et al), but they used acetic acid for hydrolysis not TFA. Although this reference is suitable for citation, the authors need to include a correct reference using TFA for the hydrolysis. The method will also need to include the incubation time used for mild acid hydrolysis.

- We agree with the reviewer that Hara et al. did not use the same hydrolysis condition. In order to comply with the reviewer, we have kept the reference to Hara et al. (now reference # 72) that reports the coupling method for the first time and added another reference (Van Beselaere et al, 2012; reference # 29) for the hydrolysis protocol. Furthermore, we have included the hydrolysis time (1h) in the methods section.

Reviewer #3 (Remarks to the Author):

In the revised manuscript, the authors have conducted additional experiments to address all major concerns. The manuscript is in good shape to be published.